# Compensatory evolution drives multidrug-resistant tuberculosis in Central Asia

**Matthias Merker[1,2†]\*, Maxime Barbier[3,4†], Helen Cox[5], Jean-Philippe Rasigade[3,4,6], Silke Feuerriegel[1,2], Thomas Andreas Kohl[1,2], Roland Diel[7], Sonia Borrell[8,9], Sebastien Gagneux[8,9], Vladyslav Nikolayevskyy[10,11], Sönke Andres[12], Ulrich Nübel[13,14], Philip Supply[15,16,17,18], Thierry Wirth[3,4], Stefan Niemann[1,2]\***

[1]Molecular and Experimental Mycobacteriology, Research Center Borstel, Borstel, Germany; [2]German Center for Infection Research, Partner site Hamburg-Lübeck-Borstel-Riems, Germany; [3]Laboratoire Biologie Intégrative des Populations, Evolution Moléculaire, Ecole Pratique des Hautes Etudes, PSL University, Paris, France; [4]Institut de Systématique, Evolution, Biodiversité, UMR-CNRS 7205, Muséum National d'Histoire Naturelle, Université Pierre et Marie Curie, Ecole Pratique des Hautes Etudes, Sorbonne Universités, Paris, France; [5]Division of Medical Microbiology, Institute of Infectious Disease and Molecular Medicine, University of Cape Town, Cape Town, South Africa; [6]CIRI INSERM U1111, University of Lyon, Lyon, France; [7]Institute for Epidemiology, Schleswig-Holstein University Hospital, Kiel, Germany; [8]Department of Medical Parasitology and Infection Biology, Swiss Tropical and Public Health Institute, Basel, Switzerland; [9]University of Basel, Basel, Switzerland; [10]Imperial College London, London, United Kingdom; [11]Public Health England, London, United Kingdom; [12]Division of Mycobacteriology, National Tuberculosis Reference Laboratory, Research Center Borstel, Borstel, Germany; [13]Microbial Genome Research, Leibniz-Institut DSMZ- Deutsche Sammlung von Mikroorganismen und Zellkulturen, Braunschweig, Germany; [14]German Center for Infection Research, Braunschweig, Germany; [15]Université de Lille, CNRS UMR 8204, Inserm U1019, CHU de Lille, Institut Pasteur de Lille, Centre d'Infection et d'Immunité de Lille, Lille, France; [16]Centre National de la Recherche Scientifique, Unité Mixte de Recherche, Center for Infection and Immunity of Lille, Lille, France; [17]Center for Infection and Immunity of Lille, Université de Lille Nord de France, Lille, France; [18]Center for Infection and Immunity of Lille, Institut Pasteur de Lille, Lille, France

**\*For correspondence:**
mmerker@fz-borstel.de (MM);
stniemann@yahoo.de (SN)

†These authors contributed equally to this work

**Competing interests:** The authors declare that no competing interests exist.

**Abstract** Bacterial factors favoring the unprecedented multidrug-resistant tuberculosis (MDR-TB) epidemic in the former Soviet Union remain unclear. We utilized whole genome sequencing and Bayesian statistics to analyze the evolutionary history, temporal emergence of resistance and transmission networks of MDR *Mycobacterium tuberculosis* complex isolates from Karakalpakstan, Uzbekistan (2001–2006). One clade (termed Central Asian outbreak, CAO) dating back to 1974 (95% HPD 1969–1982) subsequently acquired resistance mediating mutations to eight anti-TB drugs. Introduction of standardized WHO-endorsed directly observed treatment, short-course in Karakalpakstan in 1998 likely selected for CAO-strains, comprising 75% of sampled MDR-TB isolates in 2005/2006. CAO-isolates were also identified in a published cohort from Russia (2008–2010). Similarly, the presence of mutations supposed to compensate bacterial fitness deficits was associated with transmission success and higher drug resistance rates. The genetic make-up of these MDR-strains threatens the success of both empirical and standardized MDR-TB therapies, including the newly WHO-endorsed short MDR-TB regimen in Uzbekistan.

DOI: https://doi.org/10.7554/eLife.38200.001

## Introduction

Multidrug-resistant tuberculosis (MDR-TB), caused by *Mycobacterium tuberculosis* complex (MTBC) strains that are resistant to the first-line drugs isoniazid and rifampicin, represent a threat to global TB control. Barely 20% of the estimated annual 480,000 new MDR-TB patients have access to adequate second-line treatment regimens. The majority of undiagnosed or ineffectively treated MDR-TB patients continue to transmit their infection and suffer high mortality (*WHO, 2016*).

Based on early observations that the acquisition of drug resistance could lead to reduced bacterial fitness (*Middlebrook and Cohn, 1953*), it was hypothesized that drug-resistant MTBC-strains had a reduced capacity to transmit, and would not widely disseminate in the general population (*Borrell and Gagneux, 2009*; *Billington et al., 1999*; *Burgos et al., 2003*; *Dye and Espinal, 2001*; *Andersson and Levin, 1999*). This optimistic scenario has been invalidated by the now abundant evidence for transmission of MDR and extensively drug-resistant MTBC-strains (XDR-TB; MDR-TB additionally resistant to at least one fluoroquinolone and one injectable aminoglycoside) in healthcare and community settings (*Borrell and Gagneux, 2009*; *Gagneux et al., 2006*; *Müller et al., 2013*; *Pym et al., 2002*; *Comas et al., 2012*). In former Soviet Union countries, which experience the highest MDR-TB rates worldwide, the expansion of drug-resistant MTBC-clones is thought to be promoted by interrupted drug supplies, inadequate implementation of regimens, lack of infection control and erratic treatment in prison settings (*Balabanova et al., 2004*; *Casali et al., 2014a*). Continued transmission is thought to be aided by the co-selection of mutations in the bacterial population that compensate for a fitness cost (e.g. growth deficit) associated particularly with the acquisition of rifampicin resistance mediating mutations (*Borrell and Gagneux, 2009*; *Andersson and Levin, 1999*; *Gagneux et al., 2006*; *Müller et al., 2013*; *Pym et al., 2002*; *Comas et al., 2012*). The compensatory mechanism for rifampicin-resistant MTBC-strains is proposed to be associated with structural changes in the RNA-polymerase subunits *RpoA*, *RpoB*, and *RpoC* that increase transcriptional activity and as a consequence enhance the growth rate (*Comas et al., 2012*). However, the impact of these bacterial genetic factors on the epidemiological success of MDR-MTBC strains and implications for current and upcoming MDR-TB treatment strategies remain unexplored.

We utilized whole-genome sequencing (WGS) to retrace the longitudinal transmission and evolution of MTBC-strains toward MDR/pre-XDR/XDR geno- and phenotypes in Karakalpakstan, Uzbekistan. In this high MDR-TB incidence setting, the proportion of MDR-TB among new TB-patients increased from 13% in 2001 to 23% in 2014 despite the local introduction of the World Health Organization (WHO) recommended DOTS strategy in 1998 and an initially limited MDR-TB treatment program in 2003 (*Cox et al., 2007*; *Ulmasova et al., 2013*). We expanded our analyses by including a WGS dataset of MDR-MTBC isolates from Samara, Russia (2008–2010) (*Casali et al., 2014a*) to investigate clonal relatedness, resistance and compensatory evolution in both settings.

## Results

### Study population and MTBC phenotypic resistance (Karakalpakstan, Uzbekistan)

Despite differences in sampling for cohort 1 (cross-sectional, 2001–2002) and cohort 2 (consecutive enrollment of MDR-TB patients, 2003–2006) (see Materials and methods), patients showed similar age, sex distributions, and proportion of residence in Nukus, the main city in Karakalpakstan (Uzbekistan) (*Appendix—table 1*). While the majority of strains from both cohorts were phenotypically resistant to additional first-line TB drugs (i.e. beyond rifampicin and isoniazid), combined resistance to all five first-line drugs was significantly greater in cohort 2 (47% in cohort 2 compared to 14% in cohort 1, p<0.0001). The same was true for resistance to the second-line injectable drug capreomycin (23% in cohort 2 compared to 2% in cohort 1, p=0.0001) (*Appendix—table 1*). This finding was surprising as the isolates from cohort two patients - who were treated with individualized second-line regimens predominately comprising ofloxacin as the fluoroquinolone and capreomycin as the second-line injectable - were all obtained before the initiation of their treatment. In addition, there was no formal

**eLife digest** Multidrug-resistant tuberculosis, often shortened to MDR-TB, is a public health crisis with close to half a million patients falling ill each year globally. Some strains of the bacterium *Mycobacterium tuberculosis*, which causes tuberculosis disease, are resistant to the two most effective drugs used to treat the infection. As a result, patients with MDR-TB require a longer treatment of up to two years, often with severe side effects and a low chance of cure. Resistant strains of the bacteria are usually weaker than drug-susceptible strains. So, for a long time, large MDR-TB epidemics were considered to be unlikely and outbreaks of MDR-TB were often regarded as locally contained phenomenona.

Recent research has shown that MDR-TB strains are often just as likely as drug-susceptible strains to be transmitted and therefore just as likely to cause large country-wide outbreaks. It has also become clear that the resistant bacteria acquire additional mutations over time to compensate for any weakness. However, a lack of detailed history of outbreaks has meant the role of the genetics of MDR-TB bacteria has not been fully understood. Without this knowledge, prevention of future outbreaks and containment of the most successful strains in areas with a high burden of disease is difficult.

To address this, Merker, Barbier et al. reconstructed the evolutionary history of MDR-TB strains obtained in 2001–2006 from an outbreak in Uzbekistan. Whole genome sequencing followed by statistical analysis highlighted one predomininant strain that likely emerged in the mid-1970s, when the country was part of the former Soviet Union. This strain has since acquired mutations that make it resistant to eight different drugs. The most successful bacterial strains found also had compensatory mutations that seem to aid their survival.

In 1998, the health authorities implemented a TB treatment program in the region without knowing the true extent of the MDR-TB outbreak at that time. Testing for drug resistance was not routinely available, and Merker, Barbier et al. saw that MDR-TB strains resistant to the drugs used spread in the study region and were later also found independently in Russia.

A lack of routine testing for drug resistance in TB remains common in many countries with high burdens of the disease. These findings emphasize the need for universal access to tests for TB drug resistance, therapies tailored for individual patients, and access to new and repurposed drugs to reduce the risk of future outbreaks of drug-resistant TB.

DOI: https://doi.org/10.7554/eLife.38200.002

MDR-TB treatment program in Karakalpakstan prior to 2003. These elements imply that the higher rate of resistance to capreomycin was attributable to infection by already resistant strains (i.e. to primary resistance).

## MTBC population structure and transmission rates

Utilizing WGS, we determined 6979 single-nucleotide polymorphisms (SNPs) plus 537 variants located in 28 genes and upstream regions associated with drug resistance and bacterial fitness (*Supplementary file 1*). The corresponding phylogeny revealed a dominant clade comprising 173/277 (62.5%) closely related isolates within MTBC lineage 2 (particularly Beijing-genotype) (*Figure 1*). This group, termed Central Asian Outbreak (CAO), showed a highly restricted genetic diversity (median pairwise distance of 21 SNPs, IQR 13–25) and was differentiated from a set of more diverse isolates by 38 specific SNPs (*Appendix—figure 1*, *Supplementary file 1* ). The proportion of CAO-isolates was similar between 2001–2002 and 2003–2004 (49% and 52%, respectively), but increased to 76% in 2005–06 (p<0.01). Over the same time periods, the proportions of other groups remained stable or decreased (*Appendix—figure 2*).

We then sized transmission networks (measured by transmission indexes, see Materials and methods) supposed to reflect human-to-human transmission over the last ~10 years based on a maximum of 10 differentiating SNPs between two isolates. Transmission rates varied, even among closely related outbreak isolates (*Figure 1*). Beijing-CAO-isolates formed particularly large transmission networks (>50 patients; *Figure 1*); 96.0% (166/173) of all Beijing-CAO isolates were associated with recent transmission (i.e. transmission index ≥1), versus 48.4% (31/64) of non-

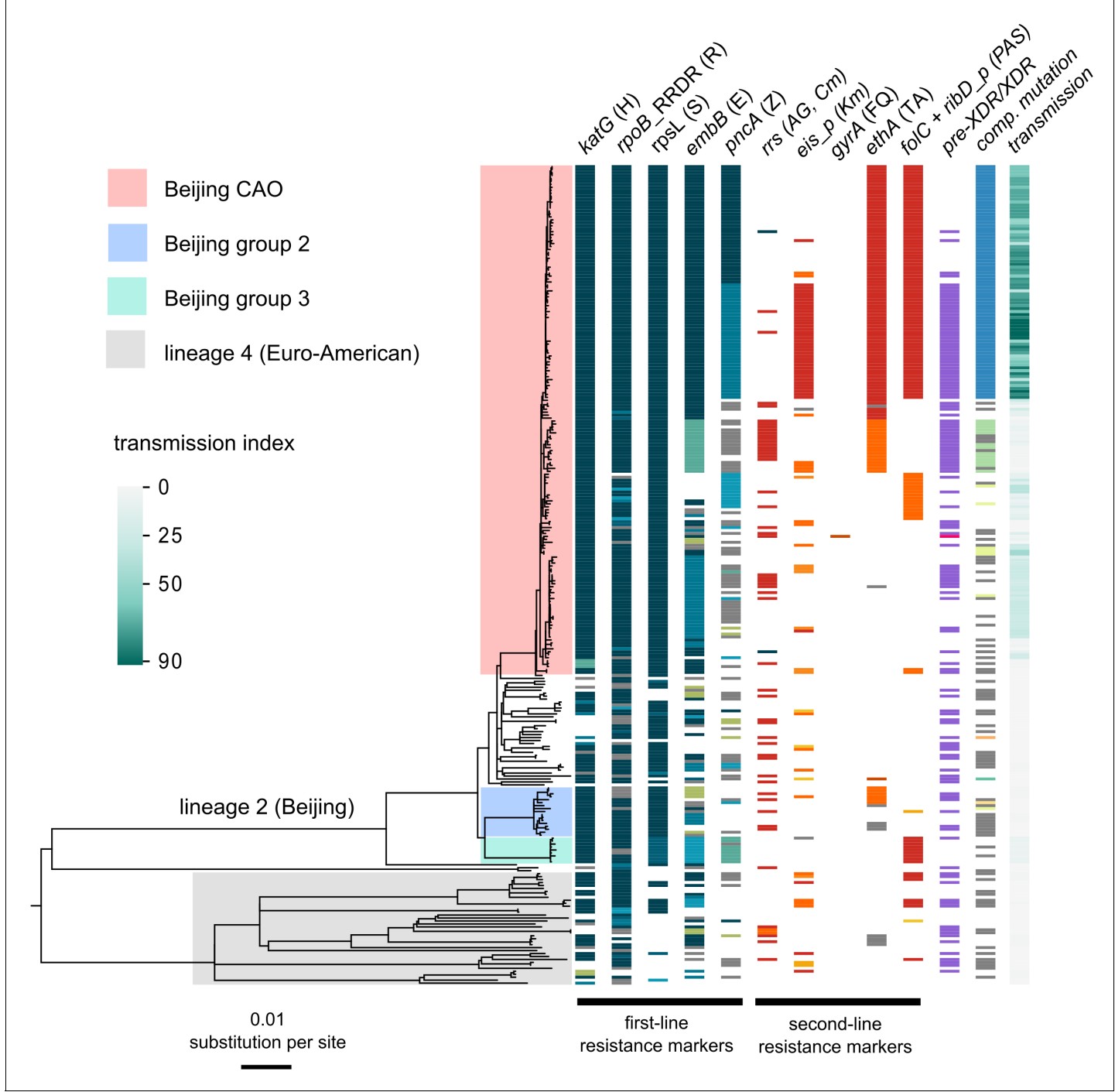

**Figure 1.** Drug resistance and transmission success among MDR-MTBC isolates from Karakalpakstan, Uzbekistan. Maximum likelihood phylogeny (GTR substitution model, 1000 resamples) of 277 MDR-MTBC isolates from Karakalpakstan, Uzbekistan sampled from 2001 to 2006. Columns show drug resistance associated mutations to first- and second-line drugs (different mutations represented by different colors), genetic classification of pre-XDR (purple) and XDR (pink) isolates, and putative compensatory mutations in the RNA polymerase genes *rpoA*, *rpoB* and *rpoC*. Transmission index represents number of isolates within a maximum range of 10 SNPs at whole genome level. MTBC lineage two isolates (Beijing genotype) are differentiated into three clades (i.e. Central Asian Outbreak (CAO), group 2 and group 3). Isolates belonging to lineage 4 (Euro-American) are colored in grey: H = isoniazid, R = rifampicin, S = streptomycin, E = ethambutol, Z = pyrazinamide, FQ = fluoroquinolone, AG = aminoglycosides, Km = kanamycin Cm = capreomycin, TA = thioamide, PAS = para aminosalicylic acid.

DOI: https://doi.org/10.7554/eLife.38200.003

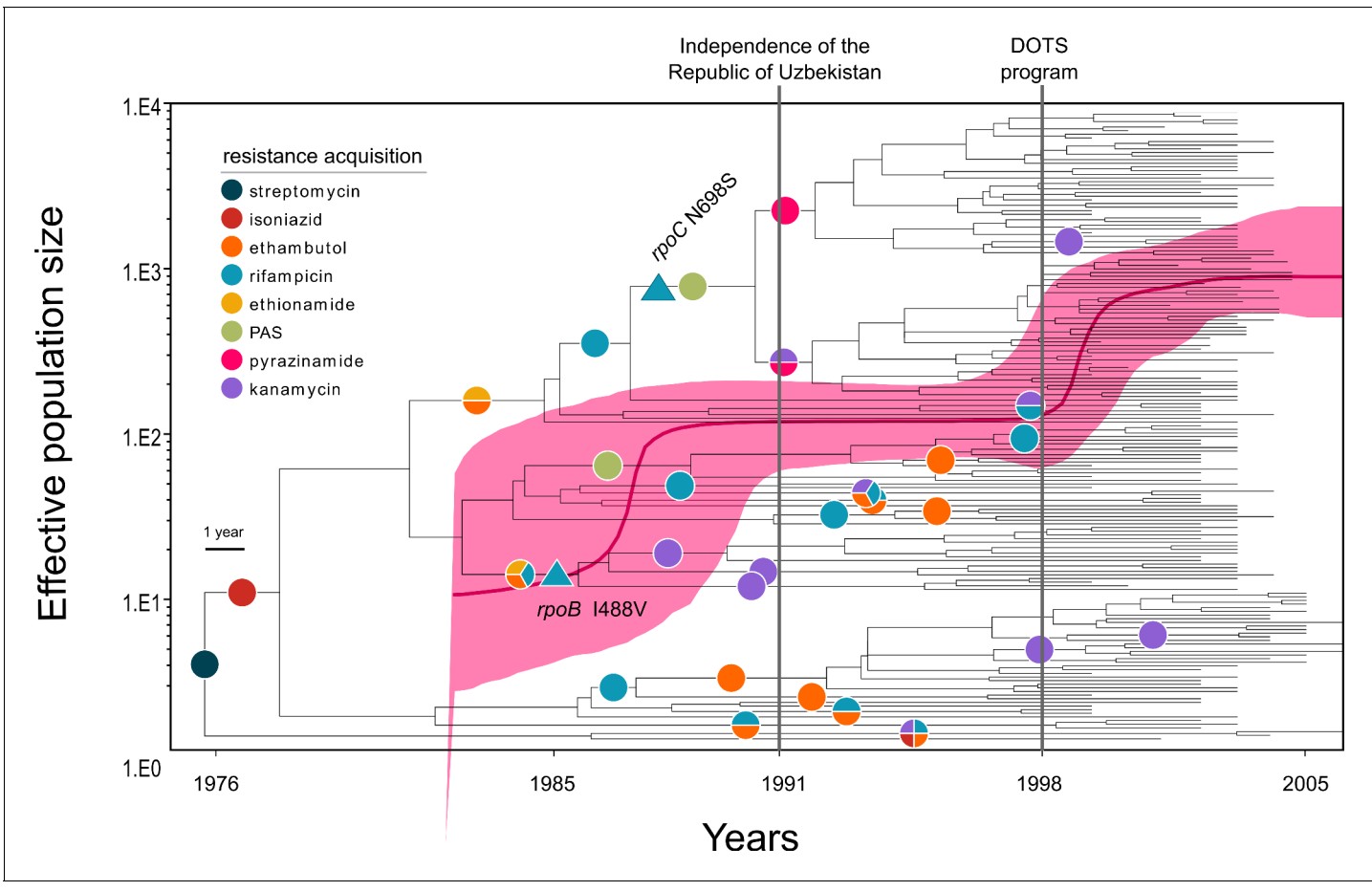

**Figure 2.** Evolutionary history of MTBC Central Asian outbreak (CAO) strains. Genealogical tree of CAO strains in Karakalpakstan, Uzbekistan and effective population size over time based on a (piecewise-constant) Bayesian skyline approach using the GTR substitution model and a strict molecular clock prior of $0.94 \times 10^{-7}$ substitutions per nucleotide per year. Pink shaded area represents changes in the effective population size giving the 95% highest posterior density (HPD) interval with the pink line representing the mean value. Vertical lines indicate time points of the implementation of the first standardized TB treatment program (DOTS) in Karakalpakstan and of the declaration of Uzbekistan as independent republic. Symbols on branches show steps of fixation of resistance conferring mutations.

DOI: https://doi.org/10.7554/eLife.38200.004

CAO Beijing isolates (p<0.0001) and 57.5% (23/40) of non-Beijing isolates (p<0.0001) (*Supplementary file 1*). In addition, the large CAO transmission network exhibited higher levels of drug resistance relative to non-Beijing strains, as reflected by the larger number of drugs for which phenotypic (p=0.0079) and genotypic drug resistance (p=0.0048) was detected *Appendix— figure 3*).

## Evolutionary history of CAO strains in Karakalpakstan

In order to gain more detailed insights into the emergence of resistance mutations in the evolutionary history of the CAO clade, we sought to employ a Bayesian phylogenetic analysis for a temporal calibration of the CAO phylogeny and an estimation of the mutation rate. Using an extended collection of more diverse CAO isolates (n = 220) from different settings (see Materials and methods), we initially compensated for the restricted sampling time frame of the Karakalpakstan dataset (2001–2006). A linear regression analysis showed correlation between sampling year and root-to-tip distance and even a moderate temporal signal (p=0.00039, $R^2$ = 5.2%, *Appendix—figure 4*), allowed for a further estimation of CAO mutation rates and evaluation of molecular clock models using Bayesian statistics as discussed previously (*Duchêne et al., 2016*). Based on the marginal L estimates collected by path sampling, we found a strict molecular clock with tip dates to be a reasonable

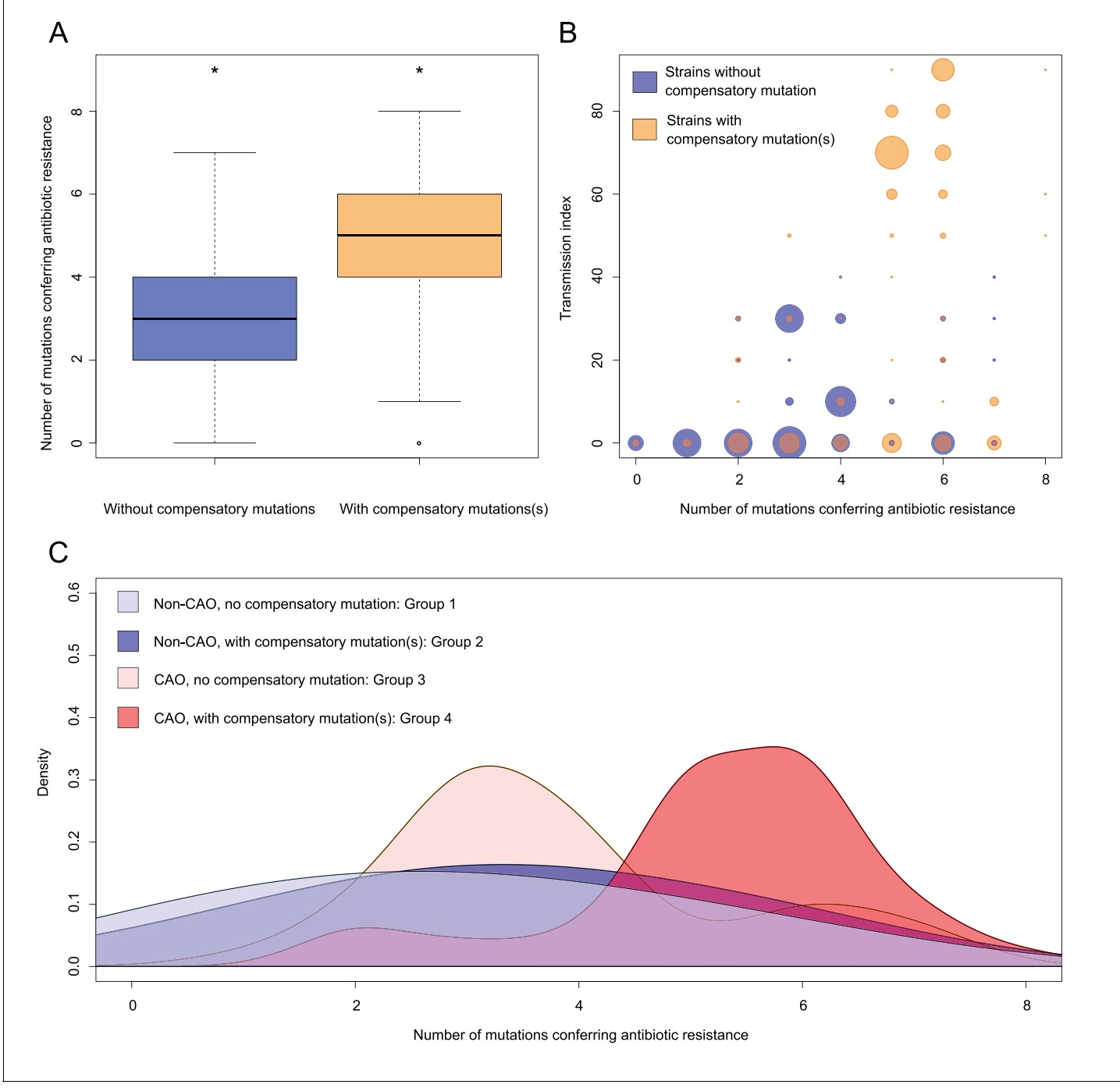

**Figure 3.** Compensatory mutations and drug resistance levels. Comparisons between isolates carrying compensatory mutations (in orange) and isolates with no-compensatory mutations (in blue), from the Karakalpakstan dataset. (**A**) Boxplot showing number of resistance mutations for the two categories (without or with compensatory mutations). The two categories were significantly different (two-sample t-test p=1.2×10$^{-10}$). (**B**) Bubble plots showing the transmission index (number of isolates differing by less than 10 SNPs) as a function of antibiotic resistance related mutations. Bubble sizes are proportional to the number of isolates. (**C**) Density plot of the number of resistance-conferring mutations for four groups of isolates sourced from the Karakalpakstan data. Proportions are adjusted by using Gaussian smoothing kernels. The four groups are composed of non-CAO isolates with no compensatory mutations; non-CAO isolates carrying compensatory mutations; CAO isolates with no compensatory mutations and CAO isolates carrying compensatory mutations. These groups are respectively colored in light blue, dark blue, light orange and light red.

DOI: https://doi.org/10.7554/eLife.38200.005

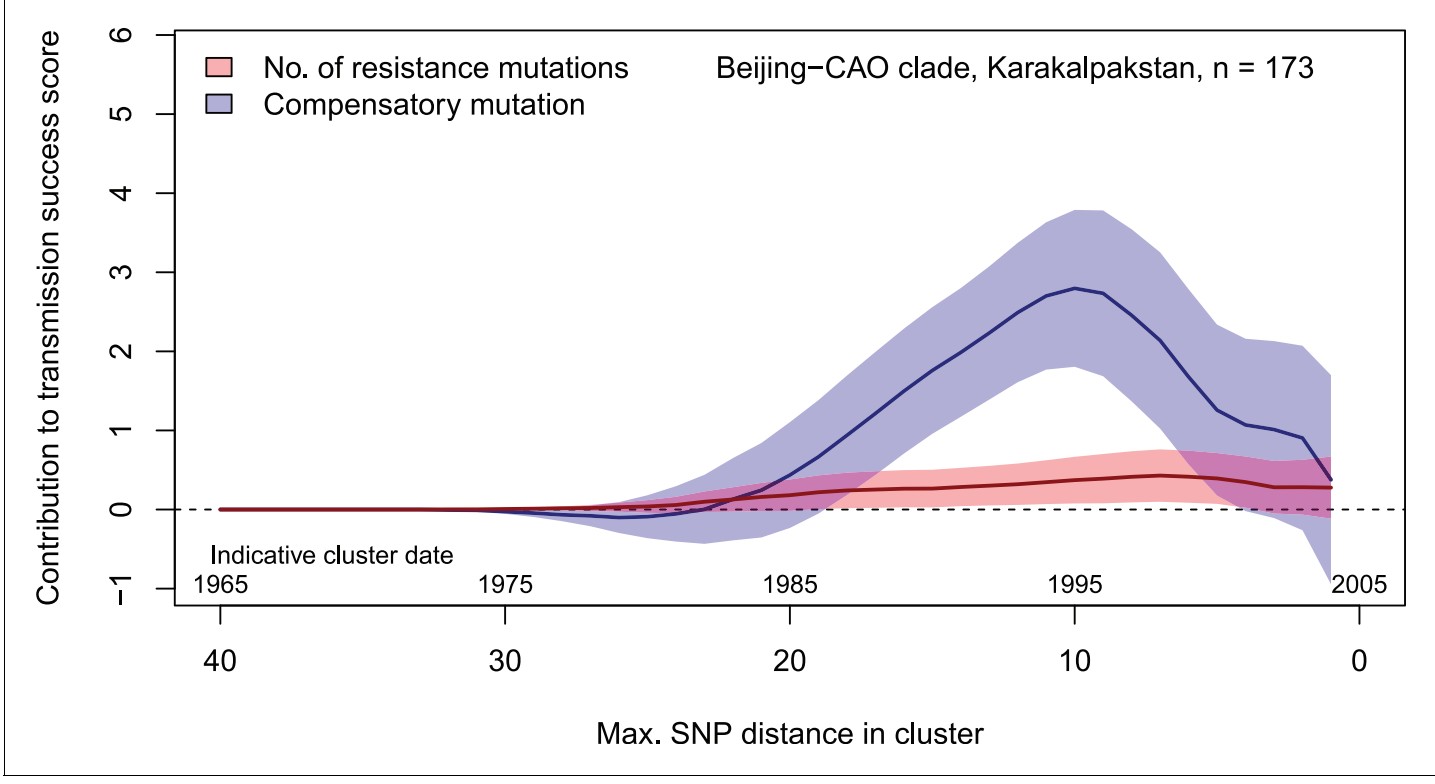

**Figure 4.** Contributions of resistance-conferring and compensatory mutations to the transmission success of the MTBC Beijing-CAO clade, Karakalpakstan, Uzbekistan. Shown are the coefficients and 95% confidence bands of multiple linear regression of the transmission success score, defined as the size of clusters diverging by at most *N* SNPs and divided by *N* or, equivalently, the size of clusters that evolved over *N* years divided by *N*. The presence of compensatory mutations was independently associated with transmission success, with a maximum association strength found for SNP distances ranging from 10 to 20 SNPs, corresponding to transmission clusters beginning around 1995.
DOI: https://doi.org/10.7554/eLife.38200.006

model for CAO isolates (*Appendix—table 2*). Mutation rate estimates (under a relaxed clock model) ranged on average from 0.88 to $0.96 \times 10^{-7}$ substitutions per site per year (s/s/y), depending on the demographic model, in favor for the Bayesian skyline model with a mutation rate of $0.94 \times 10^{-7}$ (s/s/y) (95% HPD 0.72–$1.15 \times 10^{-7}$ (s/s/y)) (*Appendix—table 2*). Comparing different demographic models for the CAO-Karakalpakstan dataset (n = 173) an exponential growth model and a Bayesian skyline model were superior over the constant size demographic prior.

Employing the Bayesian skyline model with a strict molecular clock set to $0.94 \times 10^{-7}$ (s/s/y) specifically we determined that the most recent common ancestor (MRCA) of the CAO-clade emerged around 1974 (95% highest posterior density (HPD) 1969–1982). The time to the MRCA was confirmed with the exponential growth demographic model (1977, 95% HPD 1977–1982, *Appendix— table 2*). The MRCA already exhibited a streptomycin resistance mutation (*rpsL* K43R) (*Figure 2*), and acquired isoniazid resistance (*katG* S315T) in 1977 (95% HPD 1973 – 1983). The CAO-population size then rose contemporaneously with multiple events of rifampicin, ethambutol, ethionamide, and para-aminosalicylic acid resistance acquisition in different branches (*Figure 2*). As an illustration, the most frequent CAO-clone (upper clade in *Figure 2*) acquired ethambutol and ethionamide resistance mutations (*embB* M306V, *ethA* T314I) around 1984 (95% HPD 1982–1989), and an MDR-genotype (*rpoB* S450L) around 1986 (95% HPD 1985–1992). The effective population size reached a plateau before fixation of mutations in the *ribD* promoter region (leading to para-aminosalicylic acid resistance) and *rpoC* N698S, putatively enhancing its fitness around 1990 (95% HPD 1989–1994) (*Figure 2*). Independent fixation of pyrazinamide (*pncA* Q10P and I133T) and kanamycin (*eis* −12 g/ a) resistance-associated mutations was detected in 1992 and 1991 (both with 95% HPD rounded to 1991–1996) (*Figure 2*).

To further account for uncertainties of substitution rates and thus fixation of drug resistance within the CAO-clade we ran the best models (Bayesian skyline and exponential growth) with the upper and lower HPD interval of the best clock estimate (see above). Similarly, the most recent fixation of the putative compensatory mutation *rpoC* N698S was 1994 (95% HPD 1992–1996), still years before implementation of the systematic DOTS-program in Karakalpakstan in 1998. Interestingly, the DOTS implementation coincided with a second effective population size increase (*Figure 2*). At that time, distinct CAO-clades already exhibited pre-XDR (in this context MDR plus kanamycin resistance) resistance profiles, mediating resistance to as many as eight different anti-TB drugs. Of note, only a single isolate was identified as harboring a *gyrA* mutation (A90V), associated with fluoroquinolone resistance (*Supplementary file 1*). At the end of the study period in 2006, we observed a pre-XDR rate among CAO isolates of 52.0% (90/173), compared to 35.9% (23/64) among other Beijing isolates (p=0.03) and compared to 42.5% (17/40) among non-Beijing isolates (p=0.30) (*Supplementary file 1*).

## Impact of compensatory variants on transmission networks

Overall, 62.1% (172/277) of all MDR-MTBC isolates carried putative compensatory mutations (*Comas et al., 2012*; *Casali et al., 2014a*) in *rpoA* (n = 7), *rpoC* (n = 126) and *rpoB* (n = 43) (*Supplementary file 1*). These mutations were almost completely mutually exclusive, as only 4/172 isolates harbored variants in more than one RNA polymerase-encoding gene. While mutations in *rpoA* and *rpoB* were equally distributed between Beijing-CAO isolates and other non-outbreak Beijing isolates, CAO-isolates had more *rpoC* variants (56% vs 28%, p=0.003) (*Appendix—table 3*). The mutation *rpoC* N698N accounted for 79/124 (63.7%) of CAO isolates with putative compensatory effects. The mean number of resistance mutations was higher among isolates carrying compensatory mutations (*Figure 3A*), 4.77 vs 3.35 mutations (two-sample t-test p=$1.2 \times 10^{-10}$). Notably, isolates with compensatory mutations also showed larger transmission indexes than isolates presenting no compensatory mutation, 37.16 vs 9.22 (Welch two-sample t-test p<$2.2 \times 10^{-16}$) (*Figure 3B*). CAO-isolates with compensatory mutations also had more resistance-conferring mutations than CAO-isolates lacking such mutation (ANOVA, Tukey multiple comparisons of means $P$ adj = 0.0000012). There was no difference observed for the means of resistance-conferring mutations among non-CAO isolates; compensatory mutation present vs. absent ($P$ adj = 0.1978623) (*Figure 3C*).

Regression-based analyses of transmission success scores in the Beijing-CAO clade confirmed that the presence of compensatory mutations was strongly associated with cluster sizes independent of the accumulation of resistance mutations (*Figure 4*). This pattern was mostly observed for clusters initiated in the late 1980s and the 1990s.

## Combined analysis of MDR-TB cohorts from Karakalpakstan and Samara (2001–2010)

To place our analyses in a broader phylogenetic and geographic context, we combined our Karakalpakstan genome set with previously published genomes of 428 MDR-MTBC isolates from Samara (*Casali et al., 2014a*), a Russian region located ~1700 km from Nukus, Karakalpakstan. This analysis showed that Beijing-CAO isolates accounted for the third largest clade in Samara (*Casali et al., 2014a*). Conversely, the second largest clade in Samara, termed Beijing clade B according to Casali et al (*Casali et al., 2014a*; *Casali et al., 2012*), or European/Russian W148 (*Merker et al., 2015*), was represented in Karakalpakstan by a minor clade (*Figure 5*). Considering a third Beijing clade (termed clade A) restricted to Samara (*Casali et al., 2014a*), three major Beijing outbreak clades accounted for 69.6% (491/705) of the MDR-TB cases in both regions.

The three Beijing clades (A, B, and CAO) in Samara and Karakalpakstan had more drug resistance conferring mutations (in addition to isoniazid and rifampicin resistance) with means of 5.0 (SEM 0.07), 4.2 (SEM 0.18), and 4.7 (SEM 0.11), respectively (*Appendix—figure 5*), than compared to only 3.6 (SEM 0.20) additional genotypic drug resistances (p<0.0001, p=0.0143, p<0.0001) for other Beijing isolates in both settings. Isolates belonging to other MTBC genotypes (mainly lineage four clades) were found with a mean of 2.6 (SEM 0.20) additional drug resistance mediating mutations, lower than any Beijing-associated group (p≤0.0009) (*Appendix—figure 5*).

Similar to Karakalpakstan, MDR-MTBC isolates from Samara with compensatory mutations also accumulated more resistance-associated mutations (4.57 vs 2.30 mutations per genome; two-sample

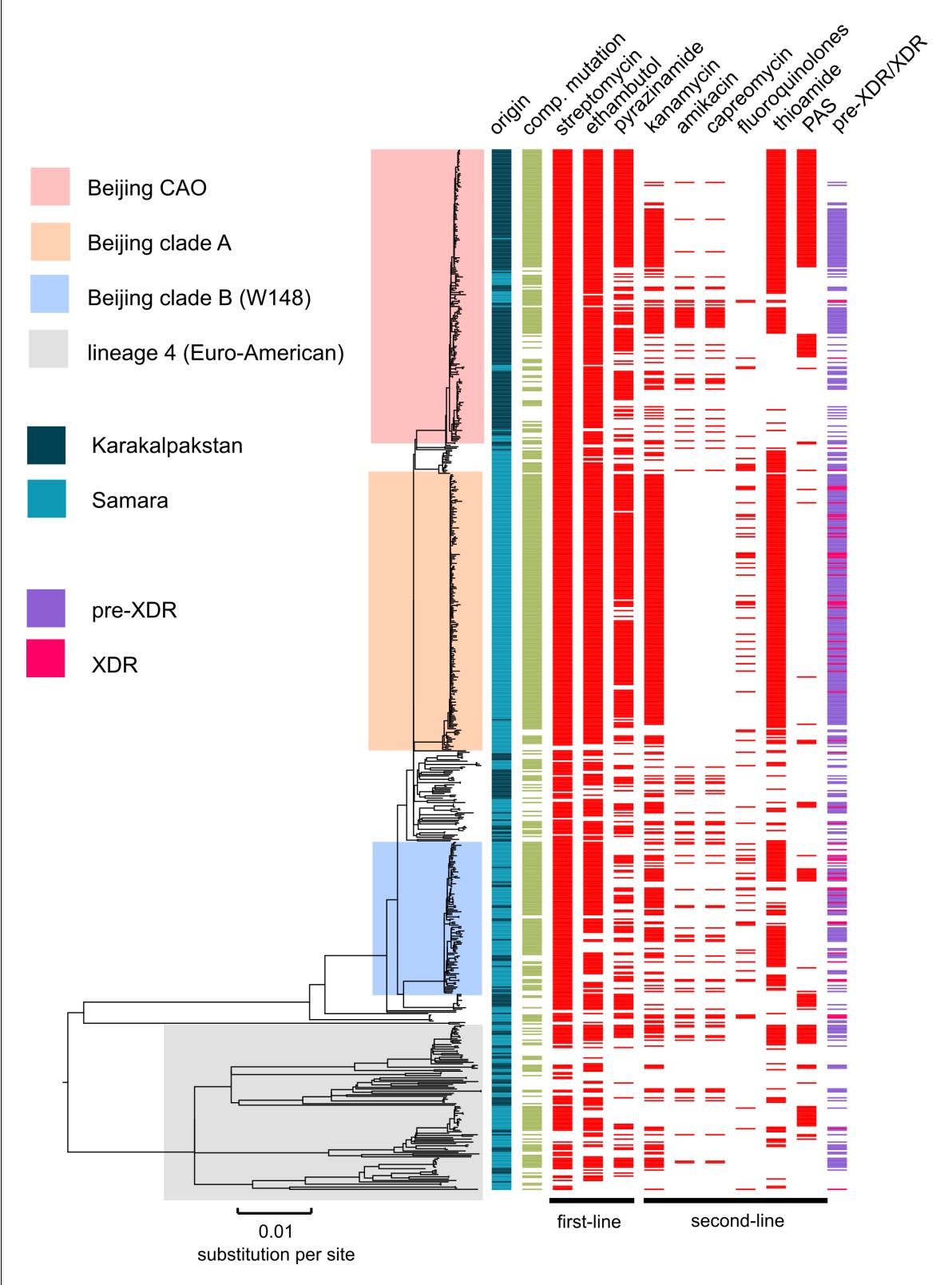

**Figure 5.** MDR-MTBC phylogeny and resistance mutations of isolates from Samara (Russia) and Karakalpakstan (Uzbekistan) Maximum likelihood tree (with 1,000 resamples, GTR nucleotide substitution model) based on 12,567 variable positions (SNPs) among 705 MDR-MTBC isolates from Karakalpakstan and Samara. Any resistance associated mutations (see methods) for individual antibiotics are depicted with red bars for each isolate. *Figure 5 continued on next page*

*Figure 5 continued*

The presence of any putative compensatory mutation in the RNA polymerase genes *rpoA*, *rpoB*, *rpoC* is depicted with green bars and country of origin and a genotypic pre-XDR and XDR isolate classification is color coded. PAS = para aminosalycylic acid.

DOI: https://doi.org/10.7554/eLife.38200.007

t-test $p<2.2\times10^{-16}$) and had higher transmission indexes (50.32 vs 0.46; Welch two-sample t-test $p<2.2\times10^{-16}$) compared to isolates lacking compensatory mutations (*Appendix—figure 6*).

The impact of resistance conferring and compensatory mutations on the transmission success score in Beijing-A clade from Samara (*Appendix—figure 7*) was strikingly similar to the one observed in CAO isolates from Karakalpakstan. The presence of compensatory mutations, but not the accumulation of resistance mutations, was significantly and independently associated with network size in clusters originating in the 1980s and 1990s, with a maximum influence found in clusters starting in the late 1990s.

Critically, the high proportions of isolates detected in both settings with pre-XDR and XDR resistance profiles among the three major Beijing clades (clade A, 96%; clade B, 62%; clade CAO, 50%; *Appendix—table 4*, *Figure 6*) reveal the low proportion of patients that are or would be eligible to receive the newly WHO endorsed short MDR-TB regimen. As per definition of the WHO exclusion criteria, for example any confirmed or suspected resistance to one drug (except isoniazid) in the short regimen, only 0.5% (1/191 in Karalalpakstan) and 2.7% (8/300 in Samara) of the patients infected with either a Beijing clade A, B or CAO strain would benefit from a shortened MDR-TB therapy (*Supplementary file 1*).

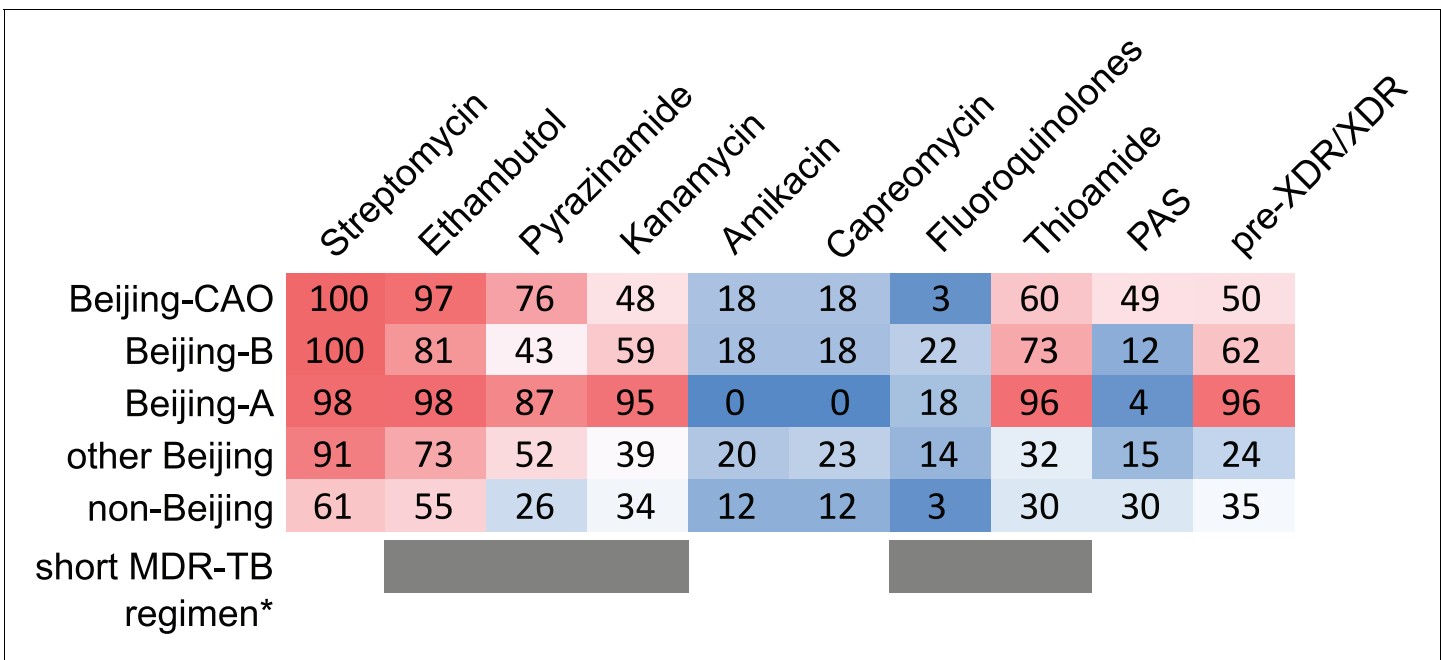

**Figure 6.** Percentage of drug resistance among 705 MDR-MTBC isolates from Samara (Russia) and Karakalpakstan (Uzbekistan). MDR-MTBC isolates stratified to three Beijing clades, other Beijing isolates and non-Beijing isolates. Proportions of isolates with identified molecular drug resistance mutations (see *Supplementary file 1*) which mediate resistance to multiple first- and second-line anti-TB drugs. Values are rounded. Drugs used in the WHO endorsed standardized short MDR-TB regimen marked with grey boxes. *The short MDR-TB regimen further includes high-dose isoniazid treatment, and clofazimine. In that regard, we identified 622/705 (85.4%) of the MDR-MTBC isolates with the well-known high-level isoniazid resistance mediating mutation *katG* S315T (*Supplementary file 1*), for clofazimine resistance mediating mutations are not well described.

DOI: https://doi.org/10.7554/eLife.38200.008

# Discussion

Using WGS combined with Bayesian and phylogenetic analyses, we reveal the evolutionary history and recent clonal expansion of the dominant MDR/pre-XDR MTBC-clade in Karakalpakstan, Uzbekistan, termed the Central Asian outbreak (CAO). Strikingly, CAO-isolates were also found also in Samara, Russia, and vice versa isolates belonging to the second largest clade in Samara (Beijing clade B, i.e. European/Russian W148 (*Casali et al., 2014a*; *Merker et al., 2015*) were identified in Karakalpakstan, suggesting that the MDR-TB epidemic in this world region is driven by few outbreak clades. During the three last decades, these strains gradually accumulated resistance to multiple anti-TB drugs that largely escaped phenotypic and molecular diagnostics, and reduced treatment options to a restricted set of drugs that often cause severe side effects. In addition, our results suggest that compensatory mutations (in RNA-polymerase subunit coding genes) that are proposed to ameliorate growth deficits in rifampicin resistant strains in vitro are also crucial in a global epidemiological context allowing MDR and pre-XDR strains to form and maintain large transmission networks. The predominance of these strain networks, seen in two distant geographic regions of the former Soviet Union clearly limit the use of standardized MDR-TB therapies, for example the newly WHO endorsed short MDR-TB regimen, in these settings.

Temporal reconstruction of the resistance mutation acquisition and of changes in bacterial population sizes over three decades demonstrates that MDR outbreak strains already became resistant to both first- and second-line drugs in the 1980s. Fully first-line resistant strains massively expanded in the 1990s, a period that shortly preceded or immediately followed the end of the Soviet Union, years before the implementation of DOTS and programmatic second-line MDR-TB treatment. This is in line with the known rise in TB incidence that accompanied the economic breakdown in Russia during the 1990s (*Institute of Medicine Forum on Drug Discovery, Development, and Translation and Russian Academy of Medical Science, 2011*).

From a bacterial genetic point of view, our data show that particular MDR and pre-XDR clades are highly transmissible despite accumulation of multiple resistance mutations. The acquisition of compensatory mutations after introduction of low fitness cost resistance mutations (e.g. *katG* S315T (*Pym et al., 2002*), *rpoB* S450L (*Gagneux et al., 2006*), *rpsL* K43R (*Böttger et al., 1998*) seems the critical stage allowing for higher transmission rates. Multiple regression analyses further strengthened this hypothesis by demonstrating that the presence of fitness compensating variants was positively associated with transmission success in different settings and outbreak clades, independently of the accumulation of resistance mutations. Compensatory evolution thus appears to play a central role in driving large MDR-TB epidemics such as that seen with the Beijing CAO-clade.

A particular concern is the high prevalence of mutations conferring resistance to second-line drugs currently included in treatment regimens, among the dominant MDR-MTBC strains. Their detected emergence in a period preceding DOTS implementation, for example in Karakalpakstan, can be explained by past, largely empirical treatment decisions or self-medication. For instance, high frequencies of mutations in the *ribD* promoter region, and *folC* among Beijing-CAO isolates, associated with para-aminosalicylic acid resistance (*Zheng et al., 2013*; *Zhao et al., 2014*), are a likely consequence of the use of para-aminosalicylic acid in failing treatment regimens in the late 1970s to the early 1980s in the Soviet Union (*USSR Ministry of Health, 1976*; *USSR Ministry of Health, 1983*; *Mishin, 2008*). Likewise, the frequent independent emergence of mutations in the *eis* promoter and of rare variants in the upstream region of *whiB7*, both linked to resistance to aminoglycosides (mainly streptomycin and kanamycin) (*Zaunbrecher et al., 2009*; *Reeves et al., 2013*), probably reflects self-administration of kanamycin that was available in local pharmacies. Of note, prominent mutations such as *katG* S315T or *rpoB* S450L might have occurred multiple times independently in a bacterial population and inferring the common ancestor could lead to an overestimate of the TMRCA. However, this is not the case for rare and more diverse mutations, for example conferring resistance to pyrazinamide, PAS or kanamycin, thus further strengthening the historic fixation mentioned above.

The pre-existence of fully first-line resistant strain populations (e.g. CAO-Beijing in Karakalpakstan) likely contributed to the poor treatment outcomes observed among MDR-TB patients following the implementation of first-line DOTS treatment in 1998 (*Cox et al., 2006*). This period coincides with a detected CAO population size increase, likely reflecting the absence of drug susceptibility

testing and therefore appropriate second-line treatment during extended hospitalization at the time, resulting in prolonged infectiousness of TB-patients and further spread of these strains.

The frequencies of fluoroquinolone resistance, mediated by *gyrA* and *gyrB* mutations, remained low among the Karakalpakstan MDR-MTBC isolates, which is consistent with the notion that such drugs were rarely used for treating TB in former Soviet Union countries (see Discussion (*Casali et al., 2014a*; *USSR Ministry of Health, 1976*; *USSR Ministry of Health, 1983*; *Mishin, 2008*). This observation explains the generally favorable MDR-TB treatment outcomes observed with the use of individualized second-line regimens, including a fluoroquinolone, in the latter MDR-TB treatment program in the Karakalpakstan patient population (*Cox et al., 2007*; *Lalor et al., 2011*). However, fluoroquinolone resistance, representing the last step towards XDR-TB, is already emerging as reported for strains in Beijing clade A and B (*Casali et al., 2014a*).

In conclusion, the (pre-) existence and wide geographic dissemination of highly resistant and highly transmissible strain populations most likely contributes to increasing M/XDR-TB incidence rates despite scaling up of the MDR-TB programs in some Eastern European and Russian regions (*Ulmasova et al., 2013*; *Institute of Medicine Forum on Drug Discovery, Development, and Translation and Russian Academy of Medical Science, 2011*; *Medecins Sans Frontiere, 2013*). Importantly, from the large spectrum of resistance detected among dominating strains in this study, it can be predicted that standardized therapies, including the newly WHO endorsed short MDR-TB regimen in Uzbekistan, are/will be largely ineffective for many patients in Samara and Karakalpakstan, and likely elsewhere in Eurasia. In order to successfully control the worldwide MDR-TB epidemics, universal access to rapid and comprehensive drug susceptibility testing, best supported by more advanced technologies, will be crucial for guiding individualized treatment with existing and new/repurposed TB drugs and to maximize chances of cure and prevention of further resistance acquisition.

## Materials and methods

### Study population, Karakalpakstan (Uzbekistan)

A total of 277 MDR-MTBC isolates derived from two separate cohorts were sequenced. The first cohort comprised 86% (49/57) of MDR-MTBC isolates from a cross-sectional drug resistance survey conducted in four districts in Karakalpakstan, Uzbekistan between 2001–2002 (*Cox et al., 2006*). An additional 228 isolates were obtained from TB-patients enrolled for second-line treatment in the MDR-TB treatment program from 2003 to 2006. These isolates represented 76% (228/300) of all MDR-TB cases diagnosed over the period. While the MDR-TB treatment program covered two of the four districts included in the initial drug resistance survey, the majority of isolates from both cohorts, 69% and 64% respectively, were obtained from patients residing in the same main city of Nukus (*Appendix—table 1*).

### Study population, Samara (Russia)

To set the MDR-MTBC isolates from Karakalpakstan into a broader geographical perspective, raw WGS data of 428 MDR-MTBC isolates from a published cross-sectional prospective study in Samara, Russia from 2008 to 2010 (*Casali et al., 2014a*) were processed as described below and included into a composite MDR-MTBC dataset.

### Drug susceptibility testing

Drug susceptibility testing (DST) was performed for five first-line drugs (isoniazid, rifampicin, ethambutol, streptomycin, pyrazinamide), and three second-line drugs (ofloxacin, capreomycin and prothionamide) for cohort 1, and six second-line drugs for cohort 2 (capreomycin, amikacin, ofloxacin, ethionamide, para-aminosalicylic acid and cycloserine) by the reference laboratory in Borstel, Germany as described previously (*Kent and Kubica, 1985*).

### Whole genome sequencing

WGS was performed with Illumina Technology (MiSeq and HiSeq 2500) using Nextera XT library preparation kits as instructed by the manufacturer (Illumina, San Diego, CA). Fastq files (raw sequencing data) were submitted to the European nucleotide archive (see *Supplementary file 1* for

accession numbers). Obtained reads were mapped to the *M. tuberculosis* H37Rv reference genome (GenBank ID: NC_000962.3) with BWA (*Li and Durbin, 2009*). Alignments were refined with GATK (*McKenna et al., 2010*) and Samtools (*Li et al., 2009*) toolkits with regard to base quality re-calibration and alignment corrections for possible PCR artefact. We considered variants that were covered by a minimum of four reads in both forward and reverse orientation, four reads calling the allele with at least a phred score of 20, and 75% allele frequency. In the combined datasets, we allowed a maximum of 5% of all samples to fail the above-mentioned threshold criteria in individual genome positions to compensate for coverage fluctuations in certain genome regions; in these cases, the majority allele was considered. Regions annotated as 'repetitive' elements (e.g. PPE and PE-PGRS gene families), insertions and deletions (InDels), and consecutive variants in a 12 bp window (putative artefacts flanking InDels) were excluded. Additionally, 28 genes associated with drug resistance and bacterial fitness (see *Supplementary file 1*) were excluded for a conservative and robust phylogenetic reconstructions. The remaining single-nucleotide polymorphisms (SNPs) were considered as valid and used for concatenated SNP alignments. Further detailed methods of the phylogenetic reconstruction, molecular resistance prediction, strain-to-strain genetic distance, and Bayesian models are given as Appendix 1.

## Transmission index

Based on the distance matrix (SNP distances), we further determined for every isolate the number of isolates that were in a range of 10 SNPs or less (in the following referred to as 'transmission index'). This 10 SNP-threshold was used to infer the number of recently linked cases, as considered within a 10-year time period, based on previous convergent estimates of MTBC genome evolution rate of ≈ 0.5 SNPs/genome/year in inter-human transmission chains and in macaque infection models (*Ford et al., 2011*; *Walker et al., 2013*; *Roetzer et al., 2013*; *Walker et al., 2014*). This can include direct transmission events among the study population but also cases which are connected by a more distant contact which was not sampled. In the latter case, we assumed that two isolates with a maximum distance of 10 SNPs share a hypothetical common ancestor that is 5 SNPs apart from the two sampled isolates (considering a bifurcating phylogeny) and thus covers a timeframe of 5 SNPs over 0.5 SNPs/year equals 10 years between the two actual samples and a shared recent ancestor node/case (see also Appendix 1).

## Genotypic drug resistance prediction

Mutations (small deletions and SNPs) in 34 resistance-associated target regions (comprising 28 genes) were considered for a molecular resistance prediction to 13 first- and second-line drugs (*Supplementary file 1*). Mutations in genes coding for the RNA-Polymerase subunits *rpoA*, *rpoB* (excluding resistance mediating mutations in the rifampicin resistance determining region (RRDR), and in codons 170, 400, 491), and *rpoC* were reported as putative fitness compensating (e.g. in vitro growth enhancing) variants for rifampicin-resistant strains as suggested previously (*Comas et al., 2011*; *de Vos et al., 2013*; *Casali et al., 2014b*; *Cohen et al., 2015*). A detailed overview of all mutations considered as genotypic resistance markers is given in *Supplementary file 1*. Mutations that were not clearly linked to phenotypic drug resistance were reported as genotypic non wild type and were not considered as genotypic resistance markers. When no mutation (or synonymous, silent mutations) was detected in any of the defined drug relevant target regions the isolate was considered to be phenotypically susceptible.

## Phylogenetic inference (maximum likelihood)

We used jModelTest v2.1 and Akaike and Bayesian Information Criterion (AIC and BIC) to find an appropriate substitution model for phylogenetic reconstructions based on the concatenated sequence alignments (*Appendix—table 5*). Maximum likelihood trees were calculated with FastTree 2.1.9 (double precision for short branch lengths) (*Price et al., 2010*) using a general time reversible (GTR) nucleotide substitution model (best model according to AIC and second best model according to BIC), 1000 resamplings and Gamma20 likelihood optimization to account for evolutionary rate heterogeneity among sites. The consensus tree was rooted with the 'midpoint root' option in Fig-Tree (resulting in the expected tree topology of lineage 2–4 strains) and nodes were arranged in increasing order. Variants considered as drug resistance markers (see above) and putative

compensatory variants were analyzed individually and mapped on the phylogenetic tree to define resistance patterns of identified phylogenetic clades.

## Molecular clock model

In order to compute a time scaled phylogeny and employ the Bayesian skyline model (see below) for the identified Central Asian outbreak (CAO) clade, we sought to define an appropriate molecular clock model (strict versus relaxed clock) and a mutation rate estimate. Due to the restricted sampling timeframe of the Karakalpakstan dataset (2001–2006), we extended the dataset for the model selection process with CAO isolates from Samara (2008–2010) and 'historical' CAO isolates from MDR-TB patients in Germany (1995–2000) thus allowing for a more confident mutation rate estimate. The strength of the temporal signal in the combined dataset, assessed by the correlation of sampling year and root-to-tip distance, was investigated with TempEst v1.5 (44). Regression analysis was based on residual mean squares, using a rooted ML tree (PhyML, GTR substitution model, 100 boot-straps), R-square and adjusted p-value are reported. For the comparison of different Bayesian phylogenetic models, we used path sampling with an alpha of 0.3, 50% burn-in and 15 million iterations (resulting in mean ESS values > 100), marginal likelihood estimates were calculated with BEAST v2.4.2 (45), and Δ marginal L estimates are reported relative to the best model.

First, we employed a strict molecular clock fixed to $1 \times 10^{-7}$ substitutions per site per year as reported previously (*Ford et al., 2011*; *Walker et al., 2013*; *Roetzer et al., 2013*) without tip dating, a strict molecular clock with tip dating and a relaxed molecular clock with tip dating. BEAST templates were created with BEAUti v2 applying a coalescent constant size demographic model, a GTR nucleotide substitution model, a chain length of 300 million (10% burn-in) and sampling of 5000 traces/trees.

Second, we ran different demographic models (i.e. coalescent constant size, exponential, and Bayesian skyline) under a relaxed molecular clock using tip dates and the same parameters for the site model and Markov-Chain-Monte-Carlo (MCMC) as described above.

Third, we tested and compared the best models for the Karakalpakstan CAO-clade under a strict molecular clock prior including the upper and lower 95% HPD interval (*Appendix—table 2*).

Inspection of BEAST log files with Tracer v1.6 showed an adequate mixing of the Markov chains and all parameters were observed with an effective sample size (ESS) >200 for the combined dataset (n = 220) and in the thousands for the Karakalpakstan CAO clade (n = 173), suggesting an adequate number of effectively independent draws from the posterior sample and thus sufficient statistical support. Other priors between the model comparisons were not changed.

## Bayesian skyline plot

Changes of the effective population size of the CAO clade in Karakalpakstan over the last four decades were calculated with a Bayesian skyline plot using BEAST v2.4.2 (45) using a tip date approach with a strict molecular clock model of $0.94 \times 10^{-7}$ substitutions per site per year (best model according to path sampling results, see above), and a GTR nucleotide substitution model. We further used a random starting tree, a chain length of 300 million (10% burn-in) and collected 5000 traces/trees. Again adequate mixing of the Markov chains and ESS values in the hundreds were observed. A maximum clade credibility genealogy was calculated with TreeAnnotator v2.

## Impact of resistance-conferring and compensatory mutations on transmission success

We used multiple linear regression to examine the respective contributions of antimicrobial resistance and putative fitness cost-compensating mutations to the transmission success of tuberculosis. To take transmission duration into account, we computed, for each isolate and each period length $T$ in years (from 1 to 40y before sampling), a transmission success score defined as the number of isolates distant of less than $T$ SNPs, divided by $T$. This approach relied on the following rationale: based on MTBC evolution rate of 0.5 mutation per genome per year, the relation between evolution time and SNP divergence is such that a cluster with at most $N$ SNPs of difference is expected to have evolved for approximately $N$ years. Thus, transmission success score over $T$ years could be interpreted as the size of the transmission network divided by its evolution time, hence as the average yearly increase of the network size. For each period $T$, the transmission success score was regressed

on the number of resistance mutations and on the presence of putative compensatory mutations. The regression coefficients with 95% confidence intervals were computed and plotted against $T$ to identify maxima, that is, time periods when the transmission success was maximally influenced by either resistance-conferring or –compensating mutations. These analyses were conducted independently on outbreak isolates of the Beijing-CAO clade in the Karakalpakstan cohort and of the Beijing-A clade in the Samara cohort.

## Statistical analyses

Differences between cohorts and numbers of sampled isolates per year category were performed using Chi-squared analysis (mid-P exact) or Fisher's exact test, while comparison of median age was performed using the Mann-Whitney test. p-Values for pairwise comparisons of groups regarding pairwise genetic distances, number of resistant DST results and number of resistance related mutations were calculated with an unpaired t-test (Welch correction) or a t-test according to the result of the variances comparison using a F-test. Boxplot, bubble plots and density plots have been performed in R.

## Acknowledgements

We thank: I Razio, P Vock, T Ubben and J Zallet from Borstel, Germany for technical assistance; the national and expatriate staff of Médecins Sans Frontières, Karakalpakstan; Dr. Atadjan and Dr. K Khamraev from the Ministry of Health (Karakalpakstan) for their support. Parts of this work have been supported by the European Union TB-PAN-NET (FP7-223681) project and the German Center for Infection Research and by the Leibniz Science Campus Evolutionary Medicine of the Lung (Evo-Lung). The funders had no role in study design, data collection and analysis, decision to publish, or preparation of the manuscript. Raw sequence data (fastq files) have been deposited at the European Nucleotide Archive (ENA) under the project number ERP000192.

## Additional information

### Funding

| Funder | Author |
|---|---|
| Leibniz Science Campus Evolutionary Medicine of the Lung | Matthias Merker Stefan Niemann |

The funders had no role in study design, data collection and interpretation, or the decision to submit the work for publication.

### Author contributions

Matthias Merker, Conceptualization, Data curation, Formal analysis, Investigation, Visualization, Methodology, Writing—original draft, Writing—review and editing; Maxime Barbier, Formal analysis, Validation, Investigation, Visualization, Methodology, Writing—review and editing; Helen Cox, Conceptualization, Formal analysis, Investigation, Writing—review and editing; Jean-Philippe Rasigade, Software, Formal analysis, Investigation, Visualization, Methodology, Writing—review and editing; Silke Feuerriegel, Formal analysis, Investigation, Writing—review and editing; Thomas Andreas Kohl, Data curation, Software, Validation, Methodology, Writing—review and editing; Roland Diel, Data curation, Formal analysis, Writing—review and editing; Sonia Borrell, Sebastien Gagneux, Vladyslav Nikolayevskyy, Writing—review and editing; Sönke Andres, Formal analysis, Methodology, Writing—review and editing; Ulrich Nübel, Methodology, Writing—review and editing; Philip Supply, Conceptualization, Investigation, Writing—original draft, Writing—review and editing; Thierry Wirth, Conceptualization, Formal analysis, Supervision, Validation, Investigation, Methodology, Writing—original draft, Writing—review and editing; Stefan Niemann, Conceptualization, Resources, Software, Supervision, Funding acquisition, Investigation, Writing—original draft, Project administration, Writing—review and editing

Author ORCIDs
Matthias Merker ⅈD http://orcid.org/0000-0003-1386-2331
Jean-Philippe Rasigade ⅈD https://orcid.org/0000-0002-8264-0452
Vladyslav Nikolayevskyy ⅈD http://orcid.org/0000-0002-9502-0332
Stefan Niemann ⅈD http://orcid.org/0000-0002-6604-0684

Decision letter and Author response
Decision letter https://doi.org/10.7554/eLife.38200.028
Author response https://doi.org/10.7554/eLife.38200.029

## Additional files

### Supplementary files

• Supplementary file 1. 34 Central Asian outbreak (CAO) specific SNPs with annotations.
DOI: https://doi.org/10.7554/eLife.38200.009

• Transparent reporting form
DOI: https://doi.org/10.7554/eLife.38200.010

### Data availability

Sequencing data have been deposited at ENA and all the accession numbers are given in Supplementary File 1 (data_Karakalp_Samara tab).

The following previously published dataset was used:

| Author(s) | Year | Dataset title | Dataset URL | Database and Identifier |
| --- | --- | --- | --- | --- |
| Nicola Casali, Vladyslav Nikolayevskyy, Yanina Balabanova, Simon R Harris, Olga Ignatyeva, Irina Kontsevaya, Jukka Corander, Josephine Bryant, Julian Parkhill, Sergey Nejentsev, Rolf D Horstmann, Timothy Brown, and Francis Drobniewski | 2014 | Discovery of sequence diversity in Mycobacterium tuberculosis (Russia collection) | https://www.ebi.ac.uk/ena/data/view/PRJEB2138 | European Nucleotide Archive, ERP000192 |

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

## Appendix 1

DOI: https://doi.org/10.7554/eLife.38200.011

## Supplementary Methods

### Transmission index

In the context of this manuscript, we determined for every isolate the number of isolates that were in a range of 10 SNPs or less (in the following referred to as 'transmission index', see figure below).

The rationale to implement a 'transmission index' was the need to link each isolate with a continuous parameter (for further analysis) which reflects the number of recently linked cases, that is the extend of a putative transmission network. These networks are better reflected with a minimum spanning tree (rather than bifurcating phylogenies) which allows the visualization of super spreaders for instance as central nodes (see figure below). Thus an isolate with a high transmission index might well be linked to a patient that infected multiple secondary cases. The benefit compared to a categorical parameter like 'clustered' and 'not clustered' is that the transmission index has the potential to indicate transmission hotspots within an outbreak scenario and is independent from a phylogenetic clade definition, which in turn would be difficult to assign due to the close genetic relationship in MTBC outbreaks and low bootstrap values for small sub-groups at the tips of a tree. The figure below illustrates the transmission index calculation per isolate. The upper left isolate for instance just has one sampled isolate within 10 SNP distance. A direct transmission event is relatively unlikely with that distance and both isolates rather share a common ancestor/linked case that was not sampled in the study period or study area. This common ancestor would be then in a four and five SNP distance, respectively that is translated to an infection event eight and ten years ago assuming an evolutionary rate of $\approx$ 0.5 SNPs/genome/year (*Ford et al., 2011*; *Walker et al., 2013*; *Roetzer et al., 2013*; *Walker et al., 2014*). The central isolate has five other sampled isolates in proximity, which might indicate a super spreader patient and/or a particularly transmissible strain.

### Genotypic drug resistance prediction

Mutations (small deletions and SNPs) in 34 resistance associated target regions (comprising 28 genes) were considered for a molecular resistance prediction to 13 first- and second-line drugs (*Supplementary file 1*). Mutations in genes coding for the RNA-Polymerase subunits *rpoA*, *rpoB* (excluding resistance mediating mutations), and *rpoC* were reported as putative fitness compensating (e.g. in vitro growth enhancing) variants for rifampicin resistant strains. A detailed overview of all mutations considered as genotypic resistance marker is given as *Supplementary file 1*. Mutations that were not clearly linked to phenotypic drug resistance were reported as genotypic non wild type and were not considered as genotypic resistance markers. When no mutation (or synonymous, silent mutations) was detected in any of the defined drug relevant target regions the isolate was considered to be phenotypically susceptible.

### Phylogenetic inference (maximum likelihood)

We used jModelTest v2.1 and Akaike and Bayesian Information Criterion (AIC and BIC) to find an appropriate substitution model for phylogenetic reconstructions based on the concatenated sequence alignments (*Appendix—table 5*). Maximum likelihood trees were calculated with FastTree 2.1.9 (double precision for short branch lengths) (*Price et al., 2010*) using a general time reversible (GTR) nucleotide substitution model (best model according to AIC and second best model according to BIC), 1000 resamplings and Gamma20 likelihood optimization to account for evolutionary rate heterogeneity among sites. The consensus tree

was rooted with the 'midpoint root' option in FigTree and nodes were arranged in increasing order. Variants considered as drug resistance marker (see above) and putative compensatory variants were analyzed individually and mapped on the phylogenetic tree to define resistance patterns of identified phylogenetic subgroups.

## Molecular clock model

In order to compute a time scaled phylogeny and employ the Bayesian skyline model (see below) for the identified Central Asian outbreak (CAO) clade we sought to define an appropriate molecular clock model (strict versus relaxed clock) and a mutation rate estimate. Due to the restricted sampling timeframe of the Karakalpakstan dataset (2001 – 2006) we extended the dataset for the model selection process with CAO strains from Samara (2008 – 2010) and 'historical' CAO strains isolated from MDR-TB patients in Germany (1995 – 2000) thus allowing for a more confident mutation rate estimate. The strength of the temporal signal in the combined dataset, assessed by the correlation of sampling year and root-to-tip distance, was investigated with TempEst v1.5 (44). Regression analysis was based on residual mean squares, using a rooted ML tree (PhyML, GTR substitution model, 100 bootstraps), R-square and adjusted $P$-value are reported. For the comparison of different Bayesian phylogenetic models we used path sampling with an alpha of 0.3, 50% burn-in and 15 million iterations (resulting in mean ESS values > 100), marginal likelihood estimates were calculated with BEAST v2.4.2 (45), and Δ marginal L estimates are reported relative to the best model.

First, we employed a strict molecular clock fixed to $1 \times 10^{-7}$ substitutions per site per year as reported previously (*Ford et al., 2011*; *Walker et al., 2013*; *Roetzer et al., 2013*) without tip dating, a strict molecular clock with tip dating and a relaxed molecular clock with tip dating. BEAST templates were created with BEAUti v2 applying a coalescent constant size demographic model, a GTR nucleotide substitution model, a chain length of 300 million (10% burn-in) and sampling of 5000 traces/trees.

Second, we ran different demographic models (i.e. coalescent constant size, exponential, and Bayesian skyline) under a relaxed molecular clock using tip dates and the same parameters for the site model and Markov-Chain-Monte-Carlo (MCMC) as described above. Inspection of BEAST log files with Tracer v1.6 showed an adequate mixing of the Markov chains and all parameters were observed with an effective sample size (ESS) in the hundreds, suggesting an adequate number of effectively independent draws from the posterior sample and thus sufficient statistical support.

## Bayesian Skyline Plot

Changes of the effective population size of the CAO clade in Karakalpakstan over the last four decades were calculated with a Bayesian skyline plot using BEAST v2.4.2 (45) using a tip date approach with a strict molecular clock model of $0.94 \times 10^{-7}$ substitutions per site per year (best model according to path sampling results, see above), and a GTR nucleotide substitution model. We further used a random starting tree, a chain length of 300 million (10% burn-in) and collected 5000 traces/trees. Again adequate mixing of the Markov chains and ESS values in the hundreds were observed. A maximum clade credibility genealogy was calculated with TreeAnnotator v2.

## Impact of resistance-conferring and compensatory mutations on transmission success

We used multiple linear regression to examine the respective contributions of antimicrobial resistance and putative fitness cost-compensating mutations to the transmission success of tuberculosis. To take transmission duration into account, we computed, for each isolate and each period length $T$ in years (from 1 to 40y before sampling), a transmission success score defined as the number of isolates distant of less than $T$ SNPs, divided by $T$. This approach relied on the following rationale: based on MTBC evolution rate of 0.5 mutation per genome

per year, the relation between evolution time and SNP divergence is such that a cluster with at most $N$ SNPs of difference is expected to have evolved for approximately $N$ years. Thus, transmission success score over $T$ years could be interpreted as the size of the transmission network divided by its evolution time, hence as the average yearly increase of the network size. For each period $T$, the transmission success score was regressed on the number of resistance mutations and on the presence of putative compensatory mutations. The regression coefficients with 95% confidence intervals were computed and plotted against $T$ to identify maxima, that is, time periods when the transmission success was maximally influenced by either resistance-conferring or –compensating mutations. These analyses were conducted independently on outbreak strains of the Beijing-CAO clade in the Karakalpakstan cohort and of the Beijing-A clade in the Samara cohort.

## Appendix Figures and Tables

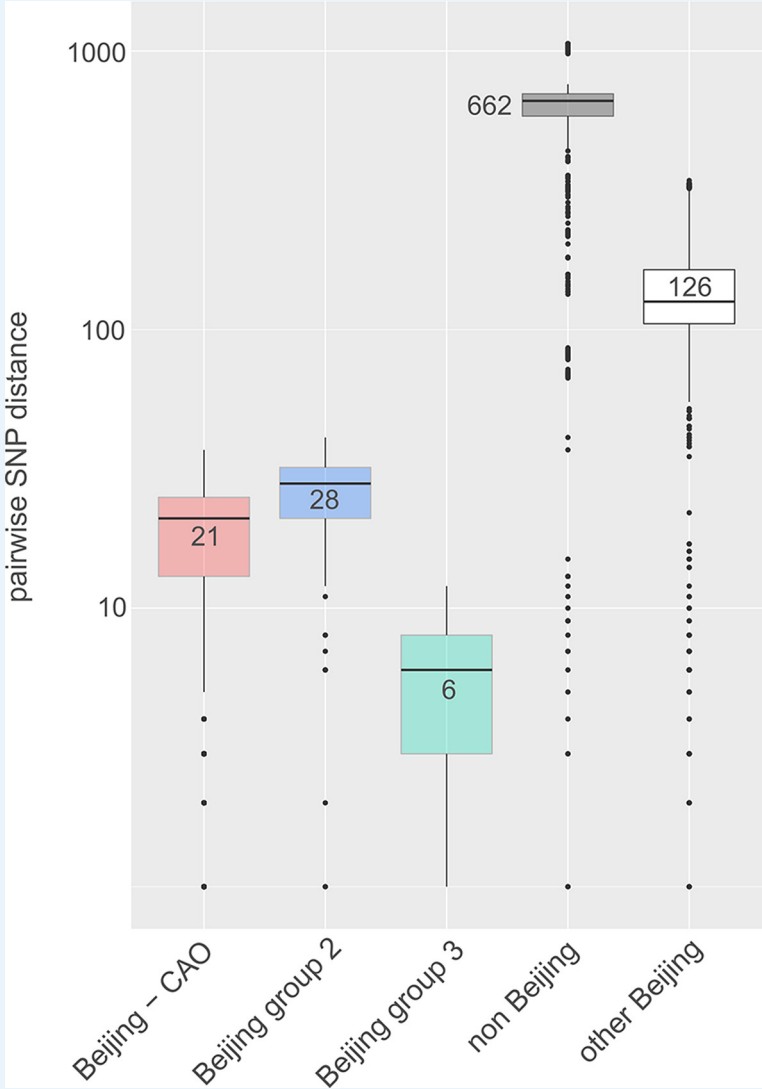

**Appendix 1—figure 1.** Box-Plot showing pairwise SNP distances among identified Beijing clades and other Beijing isolates in comparison to non-Beijing isolates from Karakalpakstan, Uzbekistan. Box represents inter quartile range, whiskers represent 95% of the data, outliers shown as black dots; solid black line represents the median.

DOI: https://doi.org/10.7554/eLife.38200.012

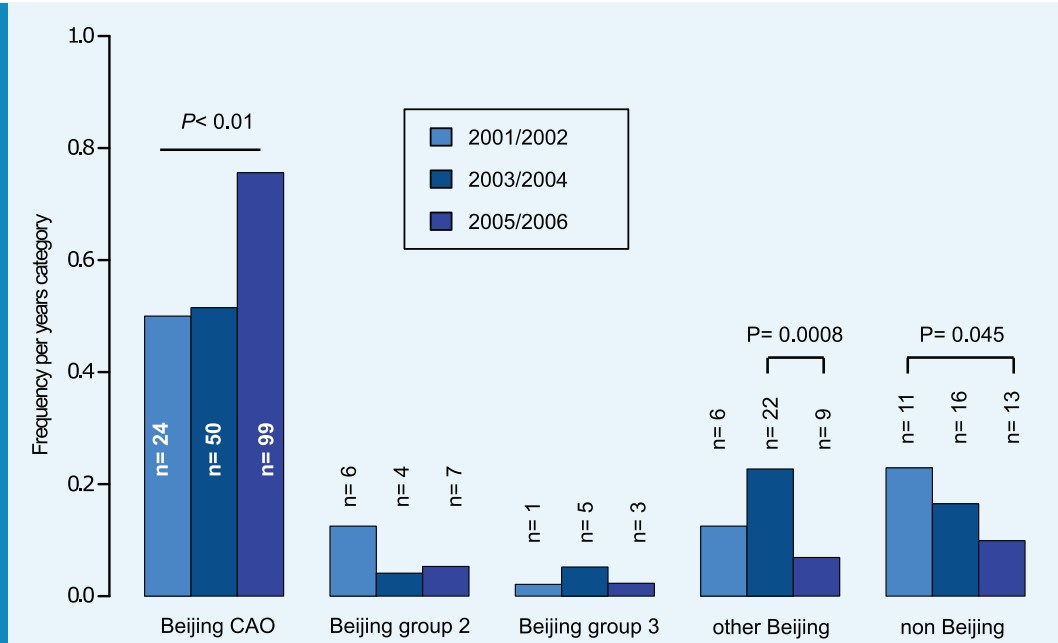

**Appendix 1—figure 2.** Proportions of different Beijing clades, other Beijing isolates and non-Beijing isolates in Karakalpakstan, Uzbekistan stratified to the years 2001/02, 2003/04, 2005/06. P-values for pairwise comparisons within groups were calculated with Fisher exact test (two-sided). Beijing CAO 2001/2002 vs 2005/2006 p=0.0018, Beijing CAO 2003/2004 vs 2005/2006 p=0.0002.

DOI: https://doi.org/10.7554/eLife.38200.013

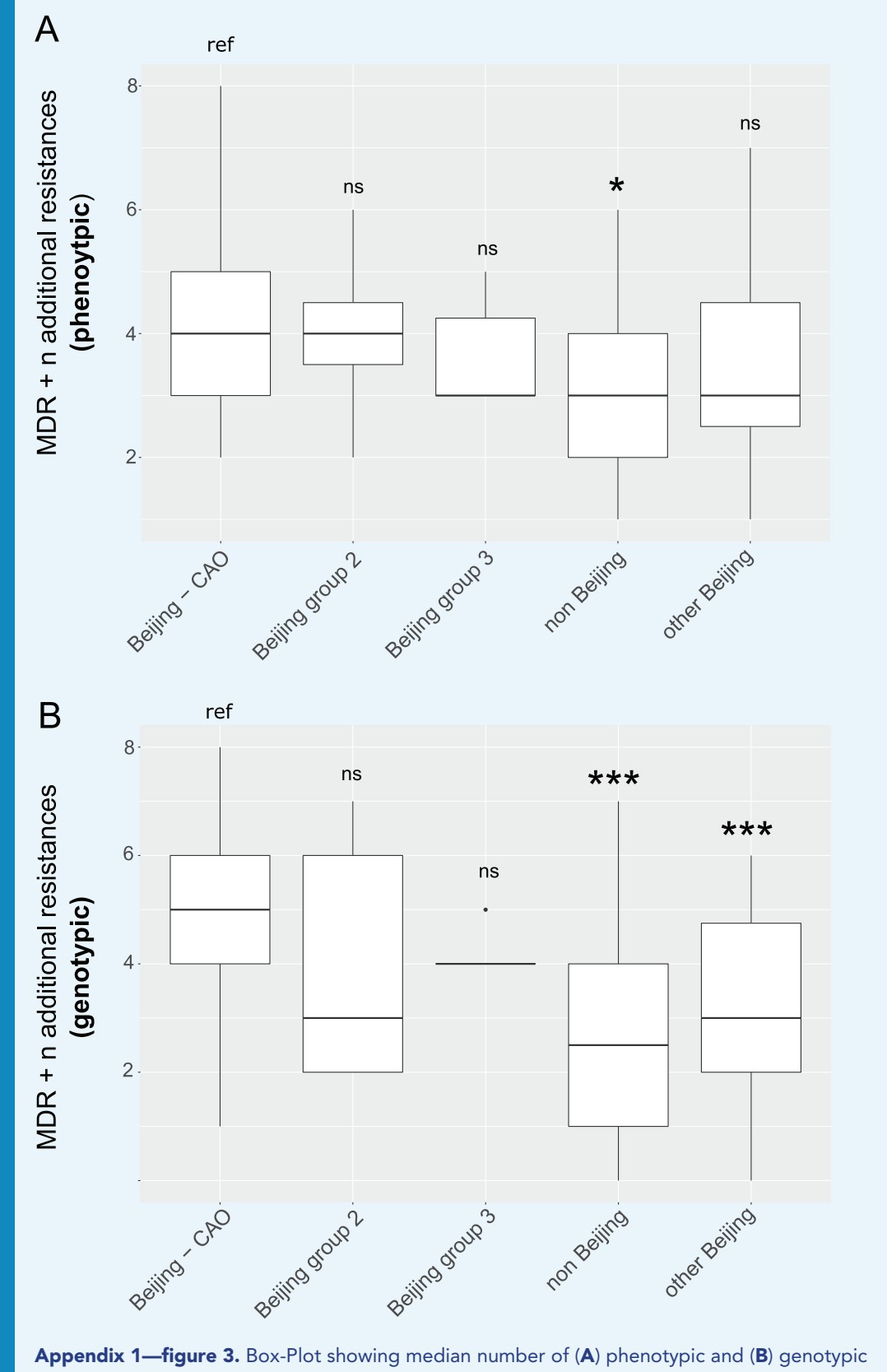

**Appendix 1—figure 3.** Box-Plot showing median number of (**A**) phenotypic and (**B**) genotypic drug resistances (in addition to the MDR classification, that is isoniazid and rifampicin resistance) of all isolates from Karakalpakstan. Box represents inter quartile range, whiskers represent 95% of the data, outliers shown as black dots; solid black line represents the median. Beijing CAO

isolates exhibit more phenotypic drug resistances compared to non-Beijing isolates (p=0.0079) and more genotypic drug resistances compared to other Beijing isolates (p<0.0001), and non-Beijing isolates (p<0.0001). P-values for pairwise comparison with reference group calculated with unpaired t-test (two-tailed, Welch's correction).

DOI: https://doi.org/10.7554/eLife.38200.014

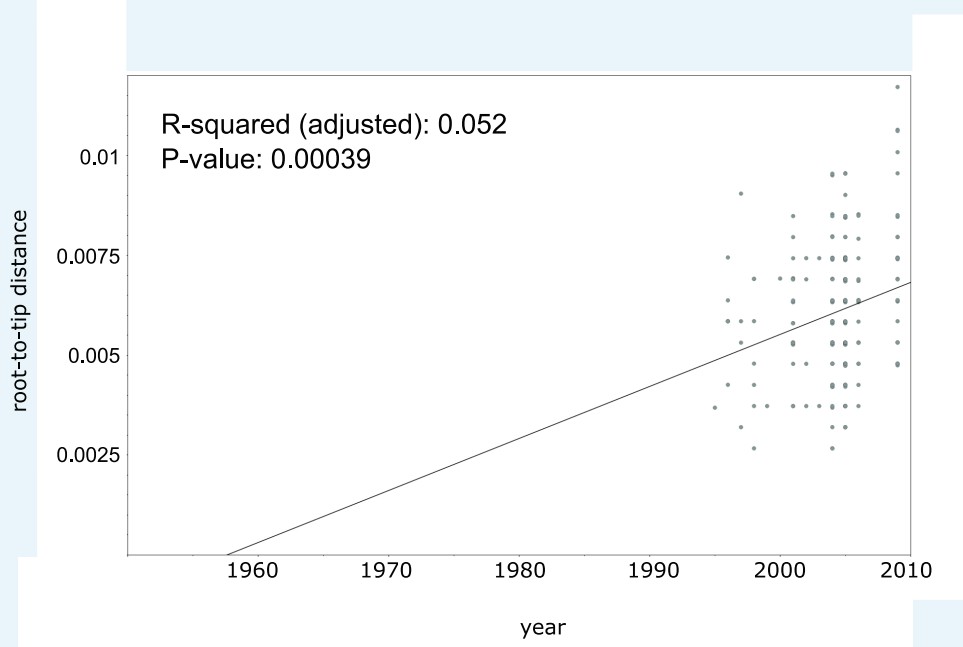

**Appendix 1—figure 4.** Linear regression analysis showing correlation between root-to-tip distance and sampling years of an extended collection of 220 Beijing CAO datasets covering the period 1995 to 2009.

DOI: https://doi.org/10.7554/eLife.38200.015

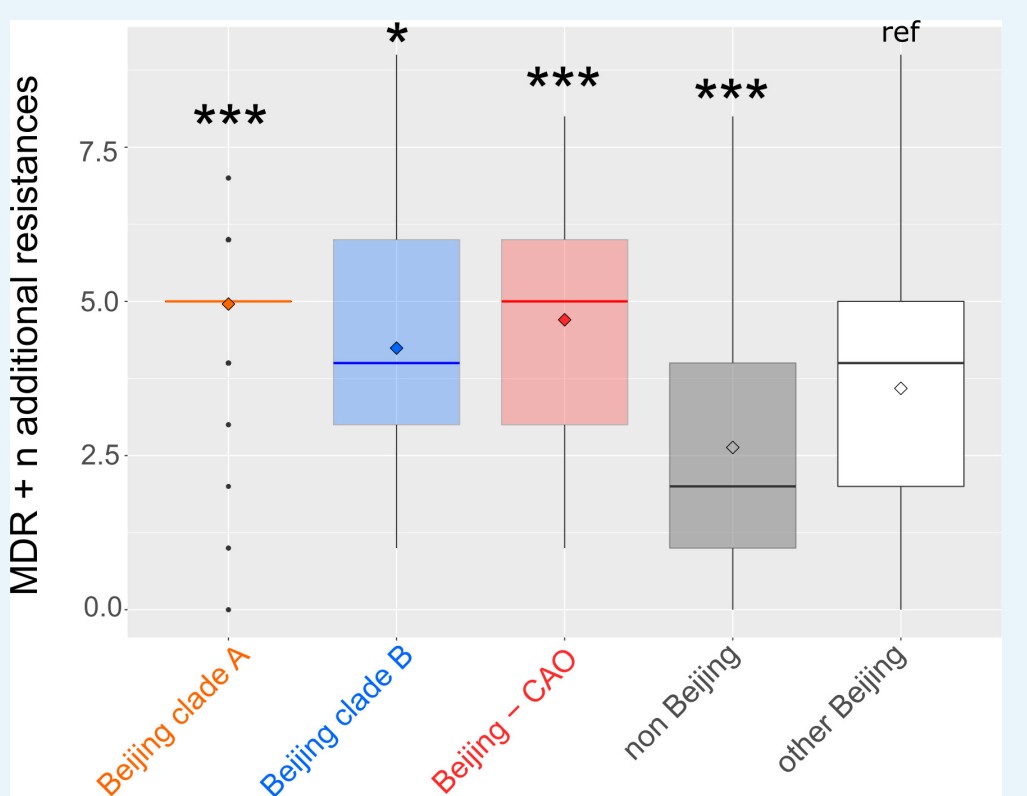

**Appendix 1—figure 5.** Number of drug resistance mutations among different MDR-MTBC groups from Samara (n = 428) and Karakapakstan (n = 277). Box-Plot with mean (diamond) and median (horizontal line) number of genotypic drug resistances (see methods) to additional anti-TB drugs (beyond MDR defining rifampicin and isoniazid resistance). Box represents inter quartile range, whiskers represent 95% of the data, outliers shown as black dots. *P*-values for three major Beijing outbreak clades (A, B and CAO), and non-Beijing isolates (mainly lineage four isolates) were calculated with unpaired t-tests with Welch correction compared to the group 'other Beijing'. Color codes according to *Figure 5*. P-values for pairwise comparison with reference group calculated with unpaired t-test (two-tailed, Welch's correction). Clade A (p≤0.0001), Clade B (p=0.0143), CAO (p≤0.0001), and non-Beijing (p=0.0009).

DOI: https://doi.org/10.7554/eLife.38200.016

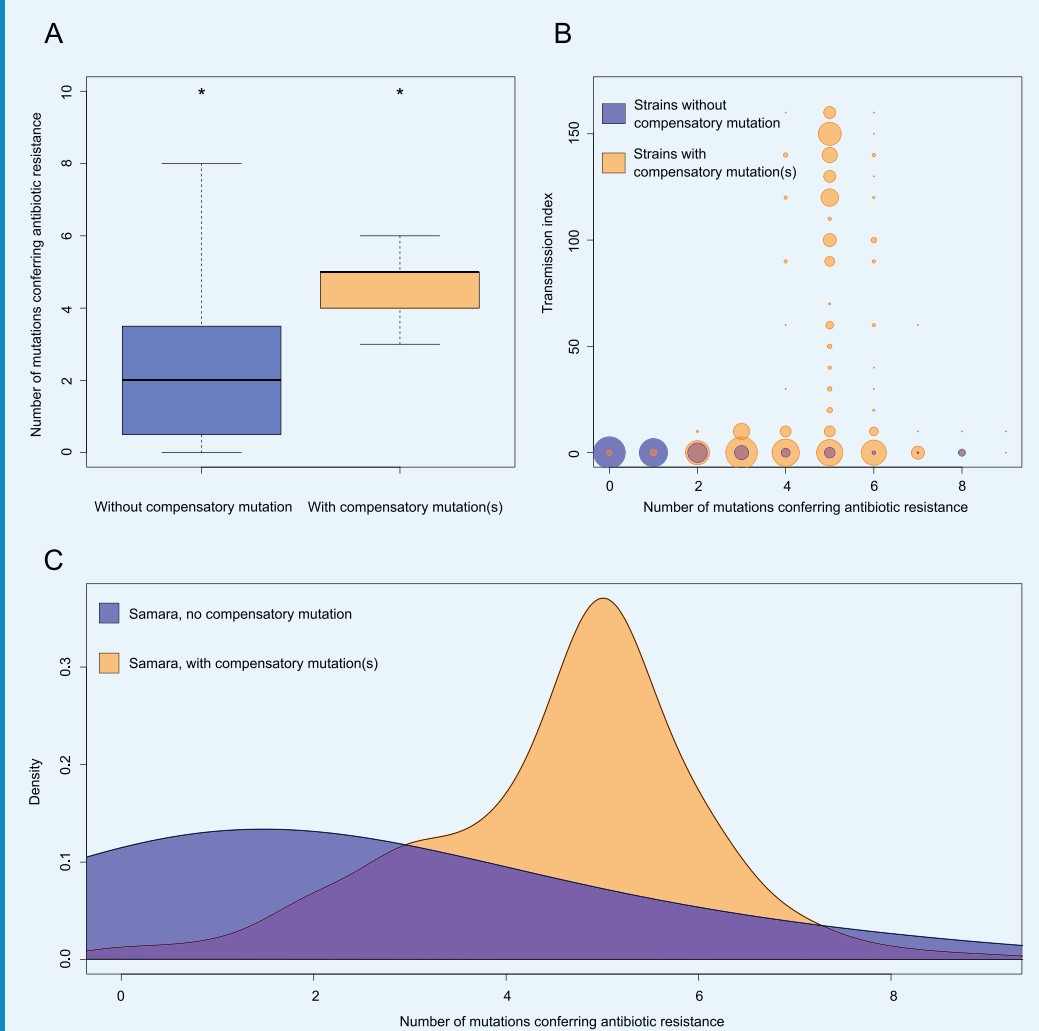

**Appendix 1—figure 6.** Comparisons between isolates carrying compensatory mutations (in orange) and isolates with no-compensatory mutations (in blue), from the Samara dataset. (**A**) Boxplot showing number of resistance mutations for the two categories (without or with compensatory mutations). The two categories were significantly different (two-sample t-test $p<2.2\times10^{-16}$). (**B**) Bubble plots showing the transmission index (number of isolates differing by less than 10 SNPs) as a function of antibiotic resistance related mutations. Bubble sizes are function of the number of isolates. (**C**) Density plot of the number of resistance-conferring mutations for isolates carrying compensatory mutations (orange) and isolates that don't carry compensatory mutation (blue) from Samara dataset. Proportions are adjusted by using Gaussian smoothing kernels.

DOI: https://doi.org/10.7554/eLife.38200.017

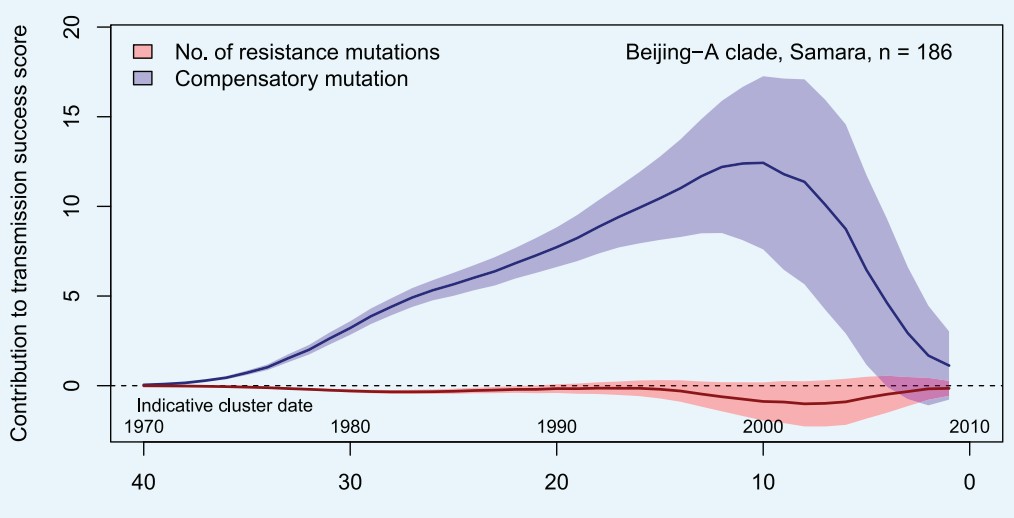

**Appendix 1—figure 7.** Contributions of resistance-conferring and compensatory mutations to the transmission success of *M.tuberculosis* of the Beijing-A clade from Samara, Russia. Shown are the coefficients and 95% confidence bands of multiple linear regression of the transmission success score, defined as the size of clusters diverging by at most *N* SNPs and divided by *N* or, equivalently, the size of clusters that evolved over *N* years divided by *N*. Compensatory mutations were independently associated with transmission success, with a maximum association strength found for transmission clusters beginning around 1999.
DOI: https://doi.org/10.7554/eLife.38200.018

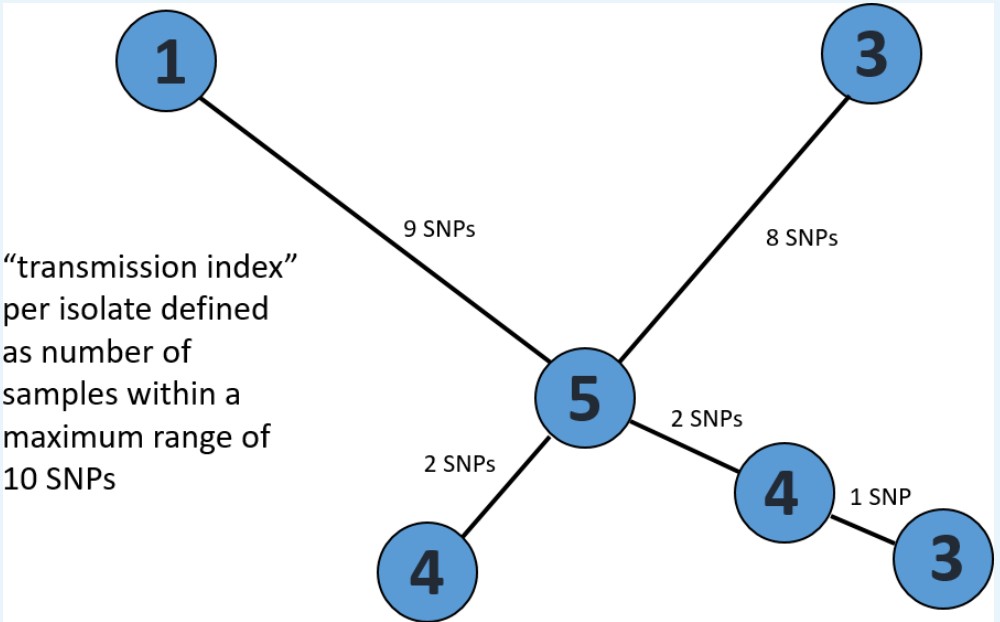

**Appendix 1—figure 8.** Exemplary minimum spanning tree to visualize the determination of a transmission index for each isolate. Each node/isolate is labelled with its transmission index, i.e. number of other isolates with a maximum distance of 10 SNPs. Branches are indicated with number of unique SNPs.
DOI: https://doi.org/10.7554/eLife.38200.019

**Appendix 1—table 1.** Main characteristics of patients from cohorts 1 and 2 in Karakalpakstan, Uzbekistan.

| | Cohort 1 | Cohort 2 | P value |
|---|---|---|---|
| Year of isolate collection (patient diagnosis with MDR-TB) | 2001–2002 | 2003–2006 | |
| No. MDR cases diagnosed within time period | 57 | 300 | |
| No. Included in this analysis | 49 (86%) | 228 (76%) | 0.094 |
| Reasons for non-inclusion: | 6 | 1 | |
| Multiple strain infection | 2 | 40 | |
| No DNA available | NA | 11 | |
| Patient already in cohort 1 | 0 | 20 | |
| Low DNA quantity | | | |
| Patient residence (within Karakalpakstan) | 34 (69%) | 146 (64%) | 0.49 |
| Nukus | 6 | 64 | |
| Chimbay | 9 | 1 | |
| Other | 0 | 17 | |
| Unknown | | | |
| Male | 27 (55%) | 119 (52%) | 0.72 |
| Age (median, IQR) | 32, 27–38 | 31, 24–41 | 0.40 |
| Missing age | 0 | 49 (21%) | |
| Previous TB treatment | 38 (78%) | 228 (100%) | <0.0001 |
| First-line resistance profile: | 1 | 2 | <0.0001 |
| HR | 0 | 1 | |
| HRE | 12 (24%) | 41 (18%) | |
| HRS | 28 (57%) | 49 (21%) | |
| HRES | 1 (2%) | 27 (12%) | |
| HRSZ | 1 | 1 | |
| HREZ | 7 (14%) | 107 (47%) | |
| HRESZ | | | |
| No. of first-line drugs resistant | 1 | 2 | <0.0001 |
| 2 | 12 (24%) | 42 (18%) | |
| 3 | 30 (61%) | 77 (34%) | |
| 4 | 7 (14%) | 107 (47%) | |
| 5 | | | |
| Availability of second-line drug susceptibility testing (DST) | Ofx, Cap, Proth | Ofx, Cap, Ami, Eth, Cyc, PAS | |
| Ofx resistance | 5 (10%) | 6 (3%) | 0.033 |
| Cm resistance | 1 (2%) | 53 (23%) | 0.0001 |

Abbreviations: H = isoniziad, R = rifampicin, E = ethambutol, S = streptomycin, Z = pyrazinamide, Ofx = ofloxacin, Cm = capreomycin

DOI: https://doi.org/10.7554/eLife.38200.020

**Appendix 1—table 2.** Path sampling results and model selection based on Δ marginal L estimates (relative to best model (ref)) considering 75 path sampling steps and chain lengths of 15 million. Comparisons of Beast runs using a combined dataset of Central Asian outbreak (CAO) isolates originated from Germany (1995 – 2000), Karakalpakstan (2001–2006), and Samara (2008 – 2010) as well as comparisons of Beast runs from CAO-Karakalpakstan data employing the best clock model/substitution rate estimate and runs with the 95% HPD intervals for the substitution rate. Mean node age of CAO-MRCA and acquisition of putative compensatory mutation rpoC N698S of CAO-Karakalpakstan clade are given for each model.

| Clock model | Demographic model | Marginal L estimate | Mean ESS | Δ marginal L estimate | Subst rate x $10^{-7}$ (95% HPD) | MRCA and rpoC N698S mean node age (95%HPD) |
|---|---|---|---|---|---|---|
| Combined CAO dataset for clock model comparison | | | | | | |

*Appendix 1—table 2 continued on next page*

*Appendix 1—table 2 continued*

| Clock model | Demographic model | Marginal L estimate | Mean ESS | Δ marginal L estimate | Subst rate x $10^{-7}$ (95% HPD) | MRCA and rpoC N698S mean node age (95%HPD) |
|---|---|---|---|---|---|---|
| Strict (no tip dating) | Coalescent constant size | −10131.67 | 4302 | 32.21 | 1.0 (fixed) | 41.5 (30.6–49.1) NA |
| Strict (tip dating) | Coalescent constant size | −10099.46 | 4041 | ref | 1.0 (fixed) | 42.9 (34.3–50.3) NA |
| Relaxed, lognormal | Coalescent constant size | −10117.21 | 1303 | 17.75 | 0.96 (0.65–1.24) | 57.6 (34.4–84.5) NA |
| Combined CAO dataset for molecular clock estimate among CAO strains | | | | | | |
| Relaxed, lognormal | Coalescent constant size | −10117.21 | 1303 | 78.28 | 0.96 (0.65–1.24) | 57.6 (34.4–84.5) NA |
| Relaxed, lognormal | Exponential | −10044.41 | 1266 | 5.48 | 0.88 (0.58–1.21) | 40.5 (26.4–53.2) NA |
| Relaxed, lognormal | Bayesian skyline | −10038.93 | 924 | ref | 0.94 (0.72–1.15) | 37.1 (25.7–44.0) NA |
| CAO-Karakalpakstan dataset for demographic model comparison (under best clock estimate) | | | | | | |
| Strict (tip dating) | Bayesian skyline | −7617.09 | 2874 | 3.79 | 0.94 (fixed) | 32.2 (23.9–37.3) 16.1 (11.6–16.9) |
| Strict (tip dating) | Coalescent constant size | −7667.92 | 4231 | 54.62 | 0.94 (fixed) | 37.5 (30.2–45.1) 15.8 (12.3–18.6) |
| Strict (tip dating) | Exponential | −7613.30 | 4003 | ref | 0.94 (fixed) | 29.3 (23.5–33.7) 15.4 (12.4–20.1) |
| CAO-Karakalpakstan dataset, exponential growth model, upper and lower 95% HPD values (from best clock estimate) | | | | | | |
| Strict (tip dating) | Exponential | −7610.94 | 3926 | ref | 0.72 (fixed) | 36.4 (30.4–43.3) 18.4 (15.8–21.6) |
| Strict (tip dating) | Exponential | −7613.30 | 4003 | 2.36 | 0.94 (fixed) | 29.3 (23.5–33.7) 15.4 (12.4–20.1) |
| Strict (tip dating) | Exponential | −7621.22 | 4031 | 10.28 | 1.15 (fixed) | 24.4 (20.0–25.8) 12.9 (10.8–14.4) |
| CAO-Karakalpakstan dataset, skyline model, upper and lower 95% HPD values (from best clock estimate) | | | | | | |
| Strict (tip dating) | Bayesian skyline | −7611.12 | 2694 | ref | 0.72 (fixed) | 39.5 (30.7–47.8) 18.8 (14.6–21.5) |
| Strict (tip dating) | Bayesian skyline | −7617.09 | 2874 | 5.97 | 0.94 (fixed) | 32.2 (23.9–37.3) 16.1 (11.6–16.9) |
| Strict (tip dating) | Bayesian skyline | −7619.71 | 2763 | 8.59 | 1.15 (fixed) | 25.2 (20.2–30.7) 11.8 (10.1–13.9) |

Abbreviations: HPD = Highest posterior density interval

DOI: https://doi.org/10.7554/eLife.38200.021

**Appendix 1—table 3.** Mutations in *rpoB*, *rpoA* and *rpoC* associated with a putative compensatory effect in rifampicin resistant MTBC strains. Data from 277 MDR-MTBC isolates from Karakalpakstan, Uzbekistan, stratified to the particularly successful variant termed Central Asian outbreak (CAO) and other Beijing isolates. Pairwise differences between the two groups calculated with Fisher exact test; two-tailed *P*-values are reported.

| | Beijing CAO (n = 173) | Other Beijing (n = 64) | P-value | All (n = 277) |
|---|---|---|---|---|
| *rpoB* mutations outside RRDR, excluding codon 170,400,491 variants<br>wild type | 25 (14.5%)<br>147 (85.0%) | 12 (18.8%)<br>52 (81.3%) | 0.43 | 43 (15.5%)<br>234 (84.5%) |
| *rpoC* variants<br>wild type | 95 (54.9%)<br>78 (45.1%) | 18 (28.1%)<br>46 (71.2%) | 0.0002 | 126 (45.5%)<br>151 (54.5%) |
| *rpoA* variants<br>wild type | 5 (2.9%)<br>168 (97.1%) | 2 (3.1%)<br>62 (96.9%) | 1.00 | 7 (2.5%)<br>270 (97.5%) |

Abbreviations: CAO = Central Asian outbreak, RRDR = rifampicin resistance determining region

DOI: https://doi.org/10.7554/eLife.38200.022

**Appendix 1—table 4.** Proportions of genotypic drug resistance rates for different anti-TB drugs (beyond isoniazid and rifampicin resistance) and pre-XDR/XDR-TB classification among 705 MDR-MTBC isolates from Samara (n = 428) and Karakalpakstan (n = 277), stratified to three identified major phylogenetic clades within the Beijing genotype/lineage and to other Beijing isolates, and to non-Beijing isolates (mainly lineage 4, Euro-American).

| Group | S | E | Z | Km | Am | Cm | Fq | Thio | PAS | Pre-XDR XDR |
|---|---|---|---|---|---|---|---|---|---|---|
| Beijing CAO (n = 201) | 201/201 100.0% | 195/201 97.0% | 152/201 75.6% | 97/201 48.3% | 37/201 18.4% | 37/201 18.4% | 6/201 3.0% | 121/201 60.2% | 99/201 49.3% | 100/201 49.8% |
| Beijing clade B (W148) (n = 103) | 103/103 100.0% | 83/103 80.6% | 44/103 42.7% | 61/103 59.2% | 18/103 17.5% | 18/103 17.5% | 23/103 22.3% | 75/103 72.8% | 12/103 11.7% | 64/103 62.1% |
| Beijing clade A (n = 187) | 184/187 98.4% | 183/187 97.9% | 163/187 87.2% | 177/187 94.7% | 0/187 0.0% | 0/187 0.0% | 33/187 17.6% | 180/187 96.3% | 7/187 3.7% | 179/187 95.7% |
| Other Beijing (n = 100) | 91/100 91.0% | 73/100 73.0% | 52/100 52.0% | 39/100 39.0% | 20/100 20.0% | 23/100 23.0% | 14/100 14.0% | 32/100 32.0% | 15/100 15.0% | 45/187 24.1% |
| Non-Beijing (n = 114) | 69/114 60.5% | 63/114 55.3% | 30/114 26.3% | 39/114 34.2% | 14/114 12.3% | 14/114 12.3% | 3/114 2.6% | 34/114 29.8% | 34/114 29.8% | 40/114 35.1% |

Abbreviations: S = streptomycin, E = ethambutol, Z = pyrazinamide, Km = kanamycin, Am = amikacin, Cm = Capreomycin, Fq = fluoroquinolone, Thio = thioamide, PAS = para aminosalicylic acid

DOI: https://doi.org/10.7554/eLife.38200.023

**Appendix 1—table 5.** Likelihood scores for different substitution models calculated with Jmodeltest 2.1 and statistical model selection based on Akaike and Bayesian Information Criteration (AIC and BIC). Best model is assumed to have the lowest criteration value. Shown are the top 10 AIC models. Substitution model used for Bayesian inference marked in bold.

| Subst. model | -lnL | AIC | Δ AIC | BIC | Δ BIC |
|---|---|---|---|---|---|
| GTR | 8837.6437 | 18567.2875 | 0.0 | 21041.0025 | 7.0748 (2) |
| GTR + I | 8837.6747 | 18569.3494 | 2.0619 (2) | 21048.6109 | 14.6832 (5) |
| GTR + G | 8838.9842 | 18571.9684 | 4.6809 (3) | 21051.2299 | 17.3022 (6) |
| GTR + I + G | 8839.0077 | 18574.0153 | 6.7278 (4) | 21058.8233 | 24.8955 (8) |
| TPM1uf | 8845.426 | 18576.852 | 9.5645 (5) | 21033.9277 | 0.0 |
| TPM1uf + I | 8845.4446 | 18578.8891 | 11.6016 (6) | 21041.5113 | 7.5836 (3) |
| TPM1uf + G | 8846.7354 | 18581.4709 | 14.1834 (7) | 21044.093 | 10.1653 (4) |
| TPM1uf + I + G | 8846.7697 | 18583.5395 | 16.252 (8) | 21051.7081 | 17.7804 (7) |
| SYM | 8860.6478 | 18607.2955 | 40.008 (9) | 21064.3712 | 30.4435 (9) |
| SYM + I | 8860.6826 | 18609.3652 | 42.0777 (10) | 21071.9874 | 38.0596 (12) |

DOI: https://doi.org/10.7554/eLife.38200.024

