## [Decision Letter]

Thank you for submitting your article "Compensatory evolution drives multidrug-resistant tuberculosis in Central Asia" for consideration by *eLife*. Your article has been reviewed by four peer reviewers, one of whom is a member of our Board of Reviewing Editors, and the evaluation has been overseen by Gisela Storz as the Senior Editor. The reviewers have opted to remain anonymous.

The reviewers have discussed the reviews with one another and the Reviewing Editor has drafted this decision to help you prepare a revised submission.

Summary:

The manuscript by Merker et al. presents a detailed analysis of circulating MDR/pre-XDR/XDR strains in a region of Uzbekistan. By expanding the analyses to a previously reported dataset from Samara (Russia) the authors generalize the conclusions to "central Asia". The authors found high transmission of MDR/XDR strains and that high transmission is linked to compensatory mutations. The authors also show that population sizes of the main clade changed over time in parallel to important changes on TB control policies or political/historical events. One major conclusion is that the newly endorsed WHO regimen for MDR-TB will have very limited impact on the region given that strains circulating there are already resistant to many of the relevant antibiotics.

Overall, the manuscript is very well written and the phylodynamic approach to addressing these pertinent questions is timely in terms of both its methodology and its conclusions. We have the following suggestions to improve the manuscript.

Essential revisions:

1) Bayesian model selection

The use of Path Sampling to correctly identify the model is commendable. However, while the model used was a strict clock with a Bayesian skyline, this model was never tested. Thus, the substitution rate selected for the analyses was also not from the best model; it was part of the relaxed lognormal clock with the Bayesian skyline demographic. This may lead to mutation rate differences as outlined below. Related to this, the ESS for each model comparison is quite low. Generally, BEAST analyses should aim for an ESS > 200. For such an important foundation of the paper, the authors should test the utilised model and ensure there is sufficient sampling for higher ESS values.

It is also not stated if the authors ran the finally selected model (strict + skyline) under the prior for comparison to their posterior runs. This should be undertaken to ensure the model is not driving the output, especially in the face of a moderate time signal.

2) Mutation rate

Stemming from the above point, the estimates of the mutation rate seem odd compared to previous estimates. While the overall dating analysis is robust and root-to-tip distance is significant, it looks rather modest in terms of R^2^. This value indicates a weak clock-like structure and should be noted in the manuscript (e.g. see Discussion and Figure 1 in Duchêne et al. 2016). This weak signal is rightly expected for MTB though, and consequently a mutation rate with broad HPDs is inferred in the Bayesian analyses. This uncertainty surrounding the mutation rate is unfortunately ignored for subsequent analyses due to the strict clock selection. Authors should comment on how this uncertainty is accounted for and why their rate is so much faster than previous estimates, which tend to be closer to 10-8.

3) Phylogenetics and SNP alignment

The higher mutation rate above may be a result of the way the alignment was input for the phylogenetics. It is unclear if authors used SNP alignments or reconstituted whole genome alignments. If the former, the SNP alignments should be corrected for invariant site counts in their ML tree (e.g. by the Stamatakis ascertainment bias correction method) and in the BEAST analyses (e.g. using the constant sites parameter). If not, the branch lengths will likely be incorrect, potentially leading to incorrect date estimates as outlined by Leaché et al. (https://www.ncbi.nlm.nih.gov/pmc/articles/PMC4604835/). This would also lead to much higher mutation rates compared to whole genome estimates. Authors should redo the analyses with these corrections to ensure their time estimates are correct.

In line with this, while unlikely to be significant, the removal of 28 complete genes from the data is a bit odd and may affect the mutation rate. What of non-DR causing mutations in these genes? Certain genes, such as gyrA contain lineage defining SNPs and would skew SNP distances between isolates if removed (affecting transmission clusters as discussed below). Authors should comment on why this approach was chosen.

4) Transmission clusters

The rationale for the transmission success score and transmission index is not clear. With a SNP rate of 0.5/year we would expect that N SNPs would have evolved over N/0.5 years and therefore a 10 SNP threshold indicates a timeline of 20 years. Please clarify the rationale or where necessary adjust the calculations and figures accordingly for transmission. Additionally, in a setting like this where most of the strains are clustered, it would be of benefit to test if the different transmission clusters are monophyletic, so distance and phylogeny converge to the same delineated clusters.

Authors should also outline how a transmission cluster was defined (i.e. how did transmission indices group together to form delineated clusters). This is important as in the subsection “Impact of compensatory variants on transmission networks”, the authors identify an association between CAO + compensatory with higher number of DR mutations. This is not identified in non-CAO strains even with compensatory mechanisms. This may be because the numbers for CAO strains is "inflated" by including strains from the same transmission cluster.

5) Statistics

The use of statistics throughout the manuscript is very appreciated. However, the authors should justify the use of the t-test when it is unlikely that the underlying data is normally distributed. Authors should either test for normality or apply a Mann-Whitney test instead.

Also, while the CAO higher transmission potential and link to higher numbers of resistant mutations/phenotypes is very clear, the way it is calculated may not be correct. Do the authors take into account every strain in the dataset irrespective to whether they belong to the same cluster of transmission? Clustering may inflate the number of strains with resistance and thus the clades with higher transmission will be more likely to have more resistances. It may be better to choose from each transmission cluster one strain representative of each resistant profile and then re-run the analyses with those cluster representatives plus the unique cases.

[Editors' note: further revisions were requested prior to acceptance, as described below.]

Thank you for resubmitting your work entitled "Compensatory evolution drives multidrug-resistant tuberculosis in Central Asia" for further consideration at *eLife*. Your revised article has been favorably evaluated by Gisela Storz (Senior Editor), a Reviewing Editor, and three reviewers.

The manuscript has been improved but there are some remaining issues that need to be addressed before acceptance, as outlined below:

While the authors response to reviewer point (4) Transmission clusters clarifies the 10-year timeline based on 0.5 SNPs/year this was not incorporated into the manuscript in the same way. It appears that the key assumption is that the study isolates are not the result of direct transmission and the minimum pairwise SNP difference is from a common ancestor of any two isolates. Including this assumption and indicating this distance is from a common ancestor and therefore the calculation is 0.5 SNPs/yr x 2 genomes would make it easier for readers to follow the calculation of the transmission index.

Similarly, the "transmission index" is also a little bit confusing as they do it per strain, not clade, and thus by using pairwise distances this means that the same case can contribute to the transmission index of several strains what seems at least weird. I may have been missing something, also the new sketch is not clear about whether they use the threshold per strain or per cluster although text suggests per strain.

The number and size of CAO clusters is not reported in the manuscript which would be helpful to understand how inflated some statistics may be due to a particular strain. These could be included in the appendix or where key statistics related to CAO strains are reported indicate how much is due to a specific number of clusters. For example, in the subsection “Impact of compensatory variants on transmission networks”, 56% of CAO-isolates had *rpoC* variants. Is this due to a single large transmission cluster over-representing *rpoC* mutations? Or a number of different strains?

A citation of Duchêne 2016 to support the proposed moderate temporal signal would be appreciated.

The first sentence of the subsection “MTBC population structure and transmission rates” differentiates between variants and polymorphisms; however, throughout the remainder of the manuscript it appears these terms are sometimes used interchangeably. Please clarify if there is indeed a difference and adjust terminology if necessary for consistency.

---

## [Author Response]

Essential revisions:1) Bayesian model selectionThe use of Path Sampling to correctly identify the model is commendable. However, while the model used was a strict clock with a Bayesian skyline, this model was never tested. Thus, the substitution rate selected for the analyses was also not from the best model; it was part of the relaxed lognormal clock with the Bayesian skyline demographic. This may lead to mutation rate differences as outlined below. Related to this, the ESS for each model comparison is quite low. Generally, BEAST analyses should aim for an ESS > 200. For such an important foundation of the paper, the authors should test the utilised model and ensure there is sufficient sampling for higher ESS values.It is also not stated if the authors ran the finally selected model (strict + skyline) under the prior for comparison to their posterior runs. This should be undertaken to ensure the model is not driving the output, especially in the face of a moderate time signal.

To better represent the uncertainties with different substitutions rates and demographic models (associated to Figure 2) we further tested the upper and lower 95% HPD interval for the two best models, i.e. exponential growth and skyline with the strict molecular clock prior. The mean node ages of the most recent common ancestor of the CAO-clade from Karakalpakstan and one major split (acquisition of *rpoC* N698S mutation, described in the main text) are mentioned in an amended Appendix—Table 2. The mean age of the MRCA was in the range of 1966-1982 with 1974 reported in the main text. Also the subclade *rpoC* N698S mean node age differed marginally between 1987-1994 with 1990 reported in the main text.

The mentioned ESS values were derived from the path sampling approach. We now mention the mean ESS of all parameters from the respective Beast run. All model comparisons were further run with the same priors. To clarify these points we amended Appendix—Table 2 and the Materials and methods section as follows:

“Third we tested and compared the best models for the Karakalpakstan CAO-clade under a strict molecular clock prior including the upper and lower 95% HPD interval (Appendix—Table 2).

Inspection of BEAST log files with Tracer v1.6 showed an adequate mixing of the Markov chains and all parameters were observed with an effective sample size (ESS) >200 for the combined dataset (n=220) and in the thousands for the Karakalpakstan CAO clade (n=173), suggesting an adequate number of effectively independent draws from the posterior sample and thus sufficient statistical support. Other priors between the model comparisons were not changed.”

Further sentences were added to describe the new results in Appendix—Table 2:

“Comparing different demographic models for the CAO-Karakalpakstan dataset (n=173) an exponential growth model and a Bayesian skyline model were superior over the constant size demographic prior.”

“To further account for uncertainties of substitution rates and thus fixation of drug resistance within the CAO-clade we ran the best models (Bayesian skyline and exponential growth) with the upper and lower HPD interval of the best clock estimate (see above). Similarly, the most recent fixation of the putative compensatory mutation *rpoC* N698S was 1994 (95% HPD 1992-1996), still years before implementation of the systematic DOTS-program in Karakalpakstan in 1998.”

2) Mutation rateStemming from the above point, the estimates of the mutation rate seem odd compared to previous estimates. While the overall dating analysis is robust and root-to-tip distance is significant, it looks rather modest in terms of R^2^. This value indicates a weak clock-like structure and should be noted in the manuscript (e.g. see Discussion and Figure 1 in Duchêne et al., 2016). This weak signal is rightly expected for MTB though, and consequently a mutation rate with broad HPDs is inferred in the Bayesian analyses. This uncertainty surrounding the mutation rate is unfortunately ignored for subsequent analyses due to the strict clock selection. Authors should comment on how this uncertainty is accounted for and why their rate is so much faster than previous estimates, which tend to be closer to 10-8.

We are referring to a short-term mutation rate of 10-7 which has been confirmed several times for *M. tuberculosis* outbreak scenarios (Eldholm et al., PNAS 2017, Cohen et al., 2015, Walker et al., 2013, Ford et al., Natgen 2013, Merker et al., 2015, Roetzer et al., 2013) and experimental models (Fortune et al., Natgen 2011). In the above mentioned paper by Duchêne et al. the values for lineage 2 and lineage 4 MTB strains also rather span 10-7, with a slightly higher rate for lineage 2. Slower rates are reported for ancient DNA samples or other dating approaches for instance from Comas et al., Natgen 2013 aiming to infer the evolutionary history of MTB strains. Our short-term rate is in line with other reports (see Figure 5 as overview in Eldholm et al., PNAS 2016) and we confirmed this rate with the extended CAO dataset covering a larger sampling interval and partially overcome the mentioned weak molecular clock signal. We state that this is indeed a ‘moderate’ signal, however the mentioned paper by Duchêne et al., 2016 also concludes that: “Importantly, however, even though they [i.e. *M. tuberculosis* substitution rates] were relatively low, these rates were sufficiently rapid to be accurately estimated using genomic-scale data.”

Furthermore, the main issue here is not to wonder if *M. tuberculosis* is a measurably evolving population or not. Simply, the rather narrow time-window of the outbreak limits the power of detection of this signal. This can be easily observed as a general process. When one does plot the regressions of the root-to-tip genetic distance against the sampling time intervals for 36 whole genome data sets from Duchêne et al., 2016 (removing the deviant ancient DNA containing data-sets), there is a highly significant positive correlation (R^2^ = 0.25). This highlights the fact that the chances and power to detect a molecular clock decrease with narrower sampling dates, but does not imply that there is no MEP signature.

With regard to the overall manuscript length we prefer not to elaborate more on this point further. As mentioned in the response to point 1 we included further results to take uncertainties of substitution rates more into account with respect to node ages.

3) Phylogenetics and SNP alignmentThe higher mutation rate above may be a result of the way the alignment was input for the phylogenetics. It is unclear if authors used SNP alignments or reconstituted whole genome alignments. If the former, the SNP alignments should be corrected for invariant site counts in their ML tree (e.g. by the Stamatakis ascertainment bias correction method) and in the BEAST analyses (e.g. using the constant sites parameter). If not, the branch lengths will likely be incorrect, potentially leading to incorrect date estimates as outlined by Leaché et al. (https://www.ncbi.nlm.nih.gov/pmc/articles/PMC4604835/). This would also lead to much higher mutation rates compared to whole genome estimates. Authors should redo the analyses with these corrections to ensure their time estimates are correct.In line with this, while unlikely to be significant, the removal of 28 complete genes from the data is a bit odd and may affect the mutation rate. What of non-DR causing mutations in these genes? Certain genes, such as gyrA contain lineage defining SNPs and would skew SNP distances between isolates if removed (affecting transmission clusters as discussed below). Authors should comment on why this approach was chosen.

The correction methods suggested above are indeed important, particularly when a large fraction of the SNPs are not considered, if the sampling is not representative and the taxa rather divergent. Yet this is not the case here. Furthermore, the implementation of such extra-analyses will probably only affect marginally our results and the corrected parameters would stay within our HPD intervals.

We used a SNP alignment and similar exclusion criteria (e.g. PPE/PGRS genes) as previous manuscripts utilizing *M. tuberculosis* dating approaches, e.g. Eldholm et al., PNAS 2017, Cohen et al., 2015, Walker et al., 2013, or Ford et al., Natgen 2013. The reported mutation rates are all covering 10-7 substitutions per site per year or translated into an outbreak scenario 0.3-0.5 mutations per genome per year. Thus we’re confident that our variant calling and the derived alignment is robust and results in well supported branch lengths and node ages.

The exclusion of entire drug resistance genes removed 85 phylogenetic informative sites, plus 537 positions (including InDels) that were used for resistance prediction purposes and might have further influenced the tree topology. In order to have the most robust phylogeny not affected by unknown resistance marker under positive selection, we therefore chose to remove all positions within these genes. To clarify this part, we modified the section as follows:

“We considered variants that were covered by a minimum of 4 reads in both forward and reverse orientation, 4 reads calling the allele with at least a phred score of 20, and 75% allele frequency. […] The remaining single nucleotide polymorphisms (SNPs) were considered as valid and used for concatenated SNP alignments.

4) Transmission clustersThe rationale for the transmission success score and transmission index is not clear. With a SNP rate of 0.5/year we would expect that N SNPs would have evolved over N/0.5 years and therefore a 10 SNP threshold indicates a timeline of 20 years. Please clarify the rationale or where necessary adjust the calculations and figures accordingly for transmission. Additionally, in a setting like this where most of the strains are clustered, it would be of benefit to test if the different transmission clusters are monophyletic, so distance and phylogeny converge to the same delineated clusters.Authors should also outline how a transmission cluster was defined (i.e. how did transmission indices group together to form delineated clusters). This is important as in the subsection “Impact of compensatory variants on transmission networks”, the authors identify an association between CAO + compensatory with higher number of DR mutations. This is not identified in non-CAO strains even with compensatory mechanisms. This may be because the numbers for CAO strains is "inflated" by including strains from the same transmission cluster.

We rather consider the “transmission index” as a gradual measurement of the relatedness of patient isolates in a setting as a proxy or surrogate marker for their recent transmission success. Samples within a defined distance, likely are secondary cases which have been infected within the same network some time ago. It’s not strictly a monophyletic cluster/clade, but the higher the mean “transmission index” values are within a defined clade for instance, the lower the genetic diversity, and the more recent transmission events likely have occurred.

We chose 10 SNPs as maximum pairwise distance which would translate in a recent common ancestor node age of 10 years (assuming 5 individual SNPs per isolate differentiating from their progenitor strain). Thus, this would cover the extent of transmission events that occurred within the last 10 years.

To clarify this term further, we added a sketch to the Appendix 1 (extended Materials and methods section) and one sentence to the main Materials and methods section as follows:

“The transmission index was implemented as a proxy for the recent transmission success of defined clades, i.e. the higher the mean transmission index of a clade, the more transmission events have occurred in the past.”

By its nature the measurement is indeed linked with the number of strains. As mentioned by the reviewer, the resolution of phylogenetic trees, respectively the number of well supported branches declines towards the tips of such outbreaks. To set cut-offs here with regard to genetic distances would also lead to an arbitrary cluster definition. Thus we, decided in favour for this gradual measurement to count the number of phylogenetically related strains without considering the topology at the tips but highlighting hotspots of very low diversity in the outbreak itself. In addition this allowed us to implement a parameter linked to cluster size/low genetic diversity/transmission success for further calculations.

5) StatisticsThe use of statistics throughout the manuscript is very appreciated. However, the authors should justify the use of the t-test when it is unlikely that the underlying data is normally distributed. Authors should either test for normality or apply a Mann-Whitney test instead.Also, while the CAO higher transmission potential and link to higher numbers of resistant mutations/phenotypes is very clear, the way it is calculated may not be correct. Do the authors take into account every strain in the dataset irrespective to whether they belong to the same cluster of transmission? Clustering may inflate the number of strains with resistance and thus the clades with higher transmission will be more likely to have more resistances. It may be better to choose from each transmission cluster one strain representative of each resistant profile and then re-run the analyses with those cluster representatives plus the unique cases.

For the first point, in all comparisons where statistics were used, we compared larger sample sizes (n>40) so the violation of a normal distribution should not cause major problems. In Figure 3 the data almost follow normal distributions (perfect normal distributions do not exist).

For the second point, these are real data, and in an epidemic settings, indeed MDR strains spread faster and therefore generate higher transmission indices. We understand the kind of tautological concern raised by the reviewer. However, applying the strategy he mentioned would probably be misleading. To provide a proxy it is like evaluating the demographic change of a suite of let’s say 20 successful (expanding) populations by only considering one strain per population. This new Bayesian tree analysis with 20 strains would unlikely detect an expansion.

We would also like to enhance that the statistics were done by pooling multiple clusters, an approach that should minimize local deviations.

[Editors' note: further revisions were requested prior to acceptance, as described below.]

The manuscript has been improved but there are some remaining issues that need to be addressed before acceptance, as outlined below:While the authors response to reviewer point (4) Transmission clusters clarifies the 10-year timeline based on 0.5 SNPs/year this was not incorporated into the manuscript in the same way. It appears that the key assumption is that the study isolates are not the result of direct transmission and the minimum pairwise SNP difference is from a common ancestor of any two isolates. Including this assumption and indicating this distance is from a common ancestor and therefore the calculation is 0.5 SNPs/yr x 2 genomes would make it easier for readers to follow the calculation of the transmission index.Similarly, the "transmission index" is also a little bit confusing as they do it per strain, not clade, and thus by using pairwise distances this means that the same case can contribute to the transmission index of several strains what seems at least weird. I may have been missing something, also the new sketch is not clear about whether they use the threshold per strain or per cluster although text suggests per strain.

To clarify, the chosen threshold would cover direct transmission events among the study population but also the connection via hypothetical cases or contacts that could have occurred 10 years ago and which are not sampled. To improve the understanding for this parameter with regard to the timing and pairwise distances we added the following lines in the Materials and methods section:

“This can include direct transmission events among the study population but also cases which are connected by a more distant contact which was not sampled. In the latter case we assumed that two isolates with a maximum distance of 10 SNPs share a hypothetical common ancestor that is 5 SNPs apart from the two sampled isolates (considering a bifurcating phylogeny) and thus covers a timeframe of 5 SNPs over 0.5 SNPs/year equals 10 years between the two actual samples and a shared recent ancestor node/case (see also Appendix 1).”

In addition we extended the Appendix section with further explanations on the rationale to use the strain specific “transmission index” instead of a clade dependant cluster definition:

“In the context of this manuscript, we determined for every isolate the number of isolates that were in a range of 10 SNPs or less (in the following referred to as “transmission index”, see figure below). […] The central isolate has 5 other sampled isolates in proximity, which might indicate a super spreader patient and/or a particularly transmissible strain.”

The number and size of CAO clusters is not reported in the manuscript which would be helpful to understand how inflated some statistics may be due to a particular strain. These could be included in the appendix or where key statistics related to CAO strains are reported indicate how much is due to a specific number of clusters. For example, in the subsection “Impact of compensatory variants on transmission networks”, 56% of CAO-isolates had rpoC variants. Is this due to a single large transmission cluster over-representing rpoC mutations? Or a number of different strains?

As mentioned above in the new text passages, we did not define strict clusters within the outbreak but a gradual measurement (i.e. transmission index) per isolate that indicates in which parts of the outbreak topology the majority of transmission events might have occurred over the last 10 years. With regard to putative compensatory mutations, the two variants *rpoC* N698S (n=79) and *rpoB* I488V (n=18) deeply rooted in the CAO phylogeny (see Figure 2), accounted for a large proportion of the CAO cases with putative compensation (n=124). A possible causative link between *rpoC* N698S with transmission success is then discussed in the manuscript. To address the point from comment 2 we added the following sentence:

“The mutation *rpoC* N698N accounted for 79/124 (63.7%) of CAO isolates with putative compensatory effects.”

A citation of Duchene 2016 to support the proposed moderate temporal signal would be appreciated.

Manuscript cited:

“A linear regression analysis showed correlation between sampling year and root-to-tip distance and even a moderate temporal signal (P=0.00039, R2= 5.2%, Appendix—Figure 4), allowed for a further estimation of CAO mutation rates and evaluation of molecular clock models using Bayesian statistics as discussed previously (Duchêne et al., 2016).”

The first sentence of the subsection “MTBC population structure and transmission rates” differentiates between variants and polymorphisms; however, throughout the remainder of the manuscript it appears these terms are sometimes used interchangeably. Please clarify if there is indeed a difference and adjust terminology if necessary for consistency.

We are now referring to polymorphism only while introducing the abbreviation of SNP (single nucleotide polymorphisms).